# Low-Rank Aggregation via Optimal Right-Space Projection

## Abstract

Federated fine-tuning with LoRA enables efficient adaptation of Foundation Models to decentralized data. However, we identify a fundamental theoretical flaw in existing approaches: independently averaging low-rank factors is algebraically inconsistent with the induced full-rank updates, due to the rotational invariance of matrix factorization. We propose **PERSIA**, a low-rank aggregation framework that computes the Frobenius-optimal rank-$r$ approximation of the full-rank average without ever materializing dense matrices. PERSIA uses local Gram whitening to normalize client-specific geometric distortions, followed by a global SVD on the right signal subspace to recover a shared basis. This ensures algebraic consistency and enables a seamless extension to Personalized FL. Empirical results on the GLUE benchmarks demonstrate that PERSIA eliminates aggregation collapse, matches the performance of full-rank fine-tuning, and improves communication efficiency by enabling longer local training.

## 1. Introduction

Recent advances in Foundation Models (FM) have demonstrated exceptional performance across diverse downstream applications, benefiting from massive-scale computational and data resources (Liu et al., 2019; Brown et al., 2020; Li et al., 2023; Liu et al., 2023; 2024). However, full-rank fine-tuning of FMs remains computationally expensive and data-intensive for users with limited resources. To address this, Parameter-Efficient Fine-Tuning (PEFT) approaches, notably Low-Rank Adaptation (LoRA) (Hu et al., 2021), have emerged as practical solutions to reduce both computational and storage requirements (Xu et al., 2023). While LoRA considerably alleviates fine-tuning overhead, individuals with limited data struggle to adapt these models to their specific tasks effectively (Mao et al., 2025).

[1]Anonymous Institution, Anonymous City, Anonymous Region, Anonymous Country. Correspondence to: Anonymous Author <anon.email@domain.com>.

Preliminary work. Under review by the International Conference on Machine Learning (ICML). Do not distribute.

**Federated Learning** (FL) offers a viable solution that enables clients to collaboratively fine-tune models using distributed local datasets without compromising privacy (Tan et al., 2022). In FL, clients periodically train and upload model parameters to a central server, which aggregates and broadcasts them back to the clients (McMahan et al., 2016).

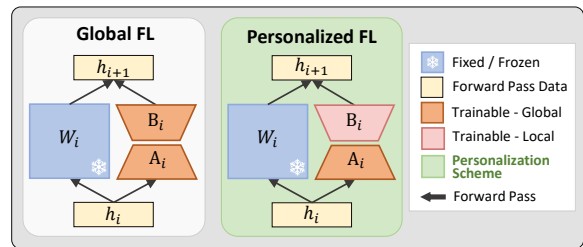

*Figure 1.* LoRA implementation in global and personalized FL.

**Challenges.** Despite LoRA's appeal for federated fine-tuning, federated aggregation of LoRA adapters is nontrivial. First, naive factor-wise averaging of LoRA parameters suffers from the *low-rank aggregation problem*:

$$\frac{1}{N}\sum_{i=1}^{N}(B_i A_i) \neq \big(\frac{1}{N}\sum_{i=1}^{N}B_i\big)\big(\frac{1}{N}\sum_{i=1}^{N}A_i\big). \quad (1)$$

The aggregated adapters deviate from the full-rank update, leading to inconsistencies between local and global steps. Second, the lack of non-linearity amplifies inconsistency, as clients can represent the same function in different bases: $BA = (BT)(T^{-1}A)$ for any invertible matrix $T \in \mathbb{R}^{r \times r}$. Third, workarounds through full-rank reconstruction have quadratic time and memory complexity that scales poorly with model size. Lastly, in personalized FL (PFL) protocols where $A$ is shared globally while $B$ remains local, the server cannot jointly access both factors to perform properly aligned aggregation, as shown in Fig. 1.

As a result, federated LoRA often resorts to restrictive local training, such as a limited number of local steps, to reduce the clients' mismatch before aggregation at the expense of increased communication frequency. These limitations highlight a fundamental gap between consistency and scalability, which motivates the following key question:

*Can we design a scalable aggregation rule for LoRA adapters that eliminates the low-rank averaging mismatch and supports substantially longer local training intervals, in both FL and PFL settings with partial parameter sharing?*

We introduce PERSIA (Principal Euclidean Right-Subspace Invariant Aggregation), a novel low-rank aggregation rule that learns a shared rank-$r$ *right subspace* by minimizing the mean-square right-projection residual across client updates. The key ideas of our method are as follows:

**(i)** We find the optimal rank-$r$ subspace spanned by the clients' right factors, and project clients' left factors into the shared subspace to ensure consistency before aggregation.

**(ii)** To ensure that the SVD recovers an optimal subspace, we apply Gram whitening on the client update to remove geometry induced by the left factors, yielding a representation whose left factor has orthonormal columns. The Gram whitening maps client updates into a common Euclidean space, where orthogonal invariance holds, and SVD recovers a Frobenius-optimal subspace.

**(iii)** In personalized FL, the whitening process happens locally, allowing a consistent aggregation of clients only based on the right factor at the server.

**(iv)** Throughout aggregation, we avoid full-rank reconstruction and dense matrix operations, reducing both time and memory complexity from quadratic to linear in the model width by operating entirely on low-rank factors.

**Contributions.** This paper investigates why federated LoRA remains communication-intensive and develops a principled, scalable remedy that applies to both global and personalized FL. We make the following contributions:

- We propose an aggregation method that mitigates the low-rank averaging problem, allows personalization of left-factors, and scales linearly with model width.

- We formalize the *LoRA aggregation error* as the mismatch between full-rank and factor-wise averaging, and showcase that PERSIA achieves the optimal Frobenius norm projection of full-rank averaging.

- We evaluate PERSIA on RoBERTa-Large across GLUE tasks. Experiments show that it can take larger local steps without collapsing, while improving convergence and reducing the communication frequency.

An overview of PERSIA is illustrated in Fig. 2. Each plane corresponds to a client-specific $r$-dimensional subspace spanned by the $r$ columns of $B_i$, denoted as $\{b_i^j\}_{j=1}^r \subset \mathbb{R}^d$. The $d$ rows of $A_i$, written as $\{a_i^k\}_{k=1}^d \subset \mathbb{R}^r$, represent coordinates within this subspace. Consequently, each parameter vector $w_i^k = \sum_{j=1}^r b_i^j a_i^k \in \mathbb{R}^d$ corresponds to mapping the coordinate $a_i^k$ from the subspace back to the ambient space.

After local training, Gram whitening is applied to remove geometry induced by the left factors. This normalization ensures that subsequent SVD of the right factors recovers a shared right factor under Euclidean geometry. By expressing left factors on a shared basis, the server can perform consistent averaging across clients.

## 2. Related work

In this section, we outline recent studies on PEFT in FL settings, with a particular focus on communication efficacy and personalization. We further discuss the alignment problem and the limitations of current LoRA solutions in the PFL, highlighting the importance of a factor-wise aggregation method that mitigates the *LoRA aggregation problem*.

**LoRA in Federated Learning.** Following LoRA's success in centralized PEFT, various studies have integrated LoRA into FL to improve communication efficiency (Fang et al., 2025), privacy preservation (Zhang et al., 2023), and personalization (Guo et al., 2024). More specifically, *personalization* takes advantage of the LoRA asymmetry, which suggests that $A$ and $B$ play distinct roles in training. The optimal choice of $A^*$ given a fix $B$ is shown to be independent of the input data distribution, implying that $A$ captures *common knowledge*, while optimal choice of $B^*$ given a fix $A$ is related to data distribution and encodes *personalized preferences* (Zhu et al., 2024). Building on this, FedSA-LoRA (Guo et al., 2024) proposes an SOTA approach that aggregates $A$ while personalizing $B$. However, aggregating LoRA adapters in FL faces a fundamental problem, as factor-wise averaging fails to capture the full-rank update that LoRA represents. This induces a functional misalignment between what clients learn and what the server broadcasts.

**LoRA Alignment.** Prior work has pursued several solutions to mitigate model-misalignment. We summarize these methods and discuss their drawbacks relative to PERSIA.

A classic approach to improving stability is to modify the local objective, such as FedProx (Sahu et al., 2020) and SCAFFOLD (Karimireddy et al., 2019). These methods are orthogonal to the LoRA averaging issue, as they reduce drift in parameter space without making LoRA factors comparable across clients. Although they improve the stability, the low-rank averaging mismatch persists.

Weight alignment is another major direction explicitly handling permutation symmetries in neural networks by matching hidden layers before averaging, including FedMA (Yurochkin et al., 2019) and subsequent weight-matching or re-basing methods such as Git Re-Basin (Ainsworth et al., 2023) and related OT-based formulations (Peyré & Cuturi, 2019). These methods significantly improve merging when models are independently trained. However, they are often expensive, and permutation alignment alone is insufficient for LoRA, where *invertible basis transformations* $BA = (BT)(T^{-1}A)$ induce $GL(T)$ symmetries that are not purely combinatorial.

Full-rank merging is a direct remedy that reconstructs and averages full-rank updates, $\frac{1}{N}\sum_i B_i A_i$, and then refactors them via SVD (Stoica et al., 2024b). This approach approximates the full-rank average well, but full-rank operations

*Figure 2.* **PERSIA Overview.** (1) Clients train locally. (2) Clients apply Gram-whitening to remove local geometric distortions (rotation/scaling) induced by the left factor $B$. (3) Server stacks all whitened factors on a plane. (4) Server performs truncated SVD to find the optimal global right subspace $A^* = V_r^\top$. (5) Clients/Server rebase the left factors $B$ into this shared coordinate system, enabling algebraically consistent averaging. (6) In Global FL, the server averages left factors $B$ and broadcasts to clients.

and decomposition dominate the cost, and it is infeasible when the server does not observe all factors (e.g., share-only-$A$ personalization) (Guo et al., 2024; Li et al., 2025). FFA-LoRA (Zhang et al., 2023) can effectively remove the LoRA averaging problem by freezing $A$ factors. However, it reduces LoRA's effective representation capacity and often yields suboptimal convergence (Guo et al., 2024).

LoRA merging methods propose merging rules that go beyond factor averaging, including sign-aware sparsified merging (Yadav et al., 2023), task-vector and arithmetic-based compositions (Ilharco et al., 2022), and LoRA-specific alignment heuristics for combining multiple adapters (Putterman et al., 2024; Zhao et al., 2024). These methods often improve practical merging, but they (i) do not provide a principled low-rank proxy for full-rank averaging, (ii) require heuristics or data-dependent selection, and (iii) do not extend to PFL protocols, where only the right factor is shared.

Taken together, existing methods typically sacrifice at least one of the following: (i) strict data-free aggregation, (ii) purely low-rank computation, (iii) provable optimality of the low-rank proxy as an approximation to full-rank averaging, (iv) compatibility with personalized protocols that share only one LoRA factor. PERSIA is designed to satisfy all four simultaneously, as we describe the methodology and provide theoretical justifications in Sections 3 and 4.

In this work, the primary empirical evidence supporting the effectiveness of PERSIA is its ability to match the performance of full-rank averaging. In addition, we include Fed-Prox, FedMA, and FFA-LoRA as complementary baselines to highlight PERSIA's advantages over existing paradigms. In personalized FL, where the server lacks access to $B$, full-rank aggregation is inapplicable, and our comparisons focus on classical FL aggregation and alignment techniques.

## 3. Methodology

This section presents the PERSIA procedure and integration into FL. We formally define the misalignment error for factor-wise averaging, and provide bounds for PERSIA for both global and personalized FL.

### 3.1. Aggregation of Principals in Low-rank via SVD

PERSIA proposes a two-stage aggregation method: first, constructing a common base, then rebasing each client solution before averaging. In this section, we describe this procedure step by step, and additionally Figs. 3 and 4 visualizes its outline for FL and PFL, and the pseudocodes are provided in Algorithms 1 and 2.

**Client Side.** In each round, clients train low-rank factors $A_i$ and $B_i$. Assuming $B_i$ has full rank, to make the low-rank geometry comparable across clients, PERSIA uses the *polar decomposition* of $B_i = O_i G_i$, where

$$G_i := (B_i^\top B_i)^{1/2} \in \mathbb{R}^{r \times r}, \qquad O_i := B_i G_i^{-1} \in \mathbb{R}^{d \times r}.$$

Therefore, $O_i^\top O_i = I_r$, and the update can be written as

$$B_i A_i = O_i(G_i A_i) = O_i Z_i, \qquad Z_i := G_i A_i \in \mathbb{R}^{r \times k}.$$

Gram-whitening converts the client-induced metric into a Euclidean one, which is the key step that makes a common right-subspace objective well-defined across clients:

$$\|\Delta W_i\|_F^2 = \mathrm{tr}(A_i^\top B_i^\top B_i A_i) = \|G_i A_i\|_F^2 = \|Z_i\|_F^2, \quad (2)$$

and for any orthogonal projector $P = P^\top = P^2 \in \mathbb{R}^{k \times k}$,

$$\|\Delta W_i(I - P)\|_F^2 = \|Z_i(I - P)\|_F^2. \qquad (3)$$

In practice, when $B_i^\top B_i$ is ill-conditioned, we compute a stabilized square root $(B_i^\top B_i + \varepsilon I_r)^{1/2}$.

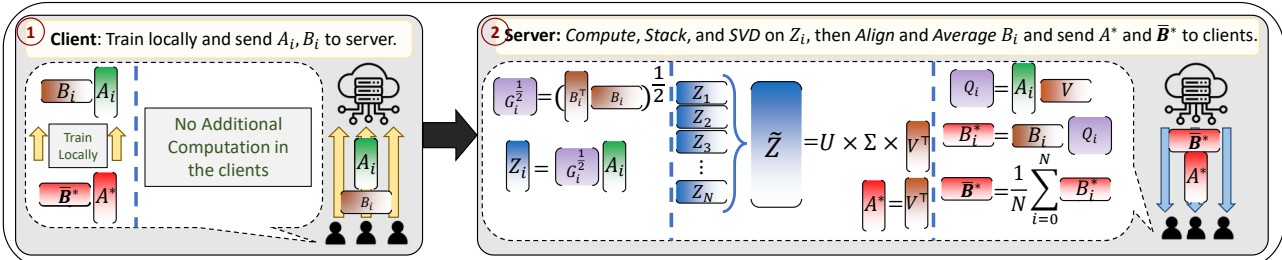

Figure 3. An overview of PERSIA steps in Global FL, where $Z_i$, $A^*$, and $B^*$ are computed at the server.

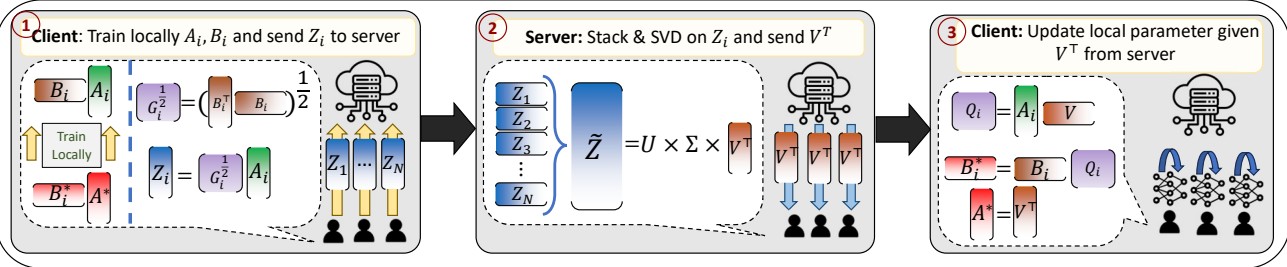

Figure 4. An overview of PERSIA steps in Personalized FL, where $Z_i$ and $B_i^*$ are computed at the clients, given $A^*$ from the server.

**Server Side in Global FL**. In global FL, the server can compute $Z_i$ from $(A_i, B_i)$, whereas in PFL (where $B_i$ is private), clients compute and communicate $Z_i$. Therefore, upon receiving (or computing) $Z_i \in \mathbb{R}^{r \times d}$, the server forms the $\tilde{Z} \in \mathbb{R}^{(Nr) \times d}$ as:

$$\tilde{Z} := \begin{bmatrix} Z_1 \\ Z_2 \\ \vdots \\ Z_N \end{bmatrix} \in \mathbb{R}^{(Nr) \times d}. \quad (4)$$

Afterwards, the server computes the top-$r$ right singular vectors $V_r \in \mathbb{R}^{d \times r}$ via truncated SVD. Equivalently, $V_r$ solves the classical principal subspace problem

$$V_r \in \operatorname*{arg\,min}_{V \in \mathbb{R}^{d \times r}:\ V^\top V = I_r} \left\| \tilde{Z} - \tilde{Z} V V^\top \right\|_F^2. \quad (5)$$

Next, to relocate the $B_i$ from their original $A_i$ base to $V_r^\top$, it computes the transformation matrix $Q_i := A_i V_r \in \mathbb{R}^{r \times r}$ for each client and updates the $B_i$ matrices as $B_i' = B_i Q_i$. Finally, the server averages the updated $B_i'$ matrices to obtain $B^\star$, which is broadcast back to clients along with $V_r^\top$ as the replacement for $A^\star$. While PERSIA alignment is based on $(O_i, Z_i)$ factorization, there is no need to realize $O_i$ and we can directly compute $B_i'$ based on $A_i$ and $V_r$,

$$B_i' = O_i(Z_i V_r) = O_i G_i A_i V_r = B_i A_i V_r = B_i Q_i. \quad (212)$$

The induced aggregated update equals a right-projection of the full-rank average:

$$B^\star A^\star = \left( \frac{1}{N} \sum_{i=0}^{N} B_i' \right) V_r^\top = \left( \frac{1}{N} \sum_{i=0}^{N} B_i A_i \right)(V_r V_r^\top), \quad (6)$$

where $P_r = (V_r V_r^\top)$ is the projection operator since:

$$(V_r V_r^\top)^2 = (V_r V_r^\top)(V_r V_r^\top) = V_r(V_r^\top V_r)V_r^\top = V_r V_r^\top.$$

**Server Side in Personalized FL**. In PFL, clients compute and upload $Z_i$ matrices. Upon receiving them, the server constructs and broadcasts $V_r^\top$ as in global FL, and the $B_i$ updates are deferred to the clients.

**Client update in Personalized FL**. After receiving the $V_r^\top$ from the server, each client relocates the $B_i$ from their original $A_i$ subspace to $V_r^\top$ via the transformation matrix $Q_i = A_i V_r \in \mathbb{R}^{r \times r}$ as $B_i^\star = B_i Q_i$. Additionally, we can show that the $B_i^\star A^\star$ is equivalent to a projection of the full-rank personalized solution into the common space:

$$B_i^\star A^\star = (B_i Q_i) V_r^\top = B_i A_i V_r V_r^\top = (B_i A_i)(V_r V_r^\top).$$

Propositions 3.3 and 3.4 show the error bounds and proves that $V_r V_r^\top$ is the optimal rank-$r$ projection in FL and PFL.

### 3.2. Optimality and aggregation error

We now formalize what PERSIA is *optimal* for, and how this yields error bounds that apply to both FL and PFL. Proofs are placed in Appendix B.

**Definition 3.1** (Aggregation error in global FL)**. *For any aggregated update $(A', B')$, define the FL aggregation error*

$$E_{\mathbf{FL}}(A', B') := \|\overline{W} - B'A'\|_F,$$

*where $\overline{W} := \frac{1}{N} \sum_{i=1}^{N} \Delta W_i \in \mathbb{R}^{d \times d}$ denotes hypothetical full-rank average (oracle aggregation).*

**Definition 3.2** (Aggregation error in PFL). *For any aggregated update $(A', B_i')$, define the PFL aggregation error*

$$E_{\mathbf{PFL}}(A', B_i') := \left( \frac{1}{N} \sum_{i=1}^{N} \| \Delta W_i - B_i' A' \|_F^2 \right)^{1/2}.$$

**Proposition 3.3** (PERSIA error in FL). *Given $\tilde{Z}$ and $V_r$ as in (4) and (5), and having $A^\star = V_r^\top$, and $B^\star = \frac{1}{N} \sum_{i=1}^{N} B_i(A_i V_r)$. Then we have*

$$E_{\mathbf{FL}}(A^\star, B^\star)^2 \leq \frac{1}{N} \| \tilde{Z} - \tilde{Z} V_r V_r^\top \|_F^2 = \frac{\mathcal{R}}{N},$$

*where $\mathcal{R} := \sum_{j>r} \sigma_j^2(\tilde{Z}) = \min_{V^\top V = I_r} \| \tilde{Z} - \tilde{Z} V V^\top \|_F^2$, denoting the stacked residual tail of $\tilde{Z}$.*

**Proposition 3.4** (PERSIA error in PFL). *Given $\tilde{Z}$ and $V_r$ as in (4) and (5), and having $A^\star = V_r^\top$, and $B^\star = B_i(A_i V_r)$. Then we have*

$$E_{\mathbf{PFL}}(A^\star, B^\star)^2 = \frac{1}{N} \| \tilde{Z} - \tilde{Z} V_r V_r^\top \|_F^2 = \frac{\mathcal{R}}{N},$$

*where $\mathcal{R} := \sum_{j>r} \sigma_j^2(\tilde{Z}) = \min_{V^\top V = I_r} \| \tilde{Z} - \tilde{Z} V V^\top \|_F^2$, denoting the stacked residual tail of $\tilde{Z}$.*

**Corollary 3.5** (Mitigate $GL(\cdot)$ Symmetry). *Suppose all clients learn the same functionality under various symmetrical representations. $W^* = W_i = BA = (BT_i)(T_i^{-1} A)$, where $T_i \in \mathbb{R}^{r \times r}$ is client-specific invertible transformation. We can show that the PERSIA factorization of $B^* A^*$ is equivalent to the ideal full-rank aggregation.*

$$E_{\mathbf{FL}}(A^*, B^*) := \| \overline{W} - B^* A^* \|_F = 0,$$

$$E_{\mathbf{PFL}}(A^*, B^*) := \left( \frac{1}{N} \sum_{i=1}^{N} \| \Delta W^* - B^* A^* \|_F^2 \right)^{1/2} = 0.$$

Propositions 3.3 and 3.4 provide exact statements that PERSIA recovers a right-projection of the oracle average $\overline{W}$, and that the resulting error is upper-bounded by (or equal to) the mean-square residual. In Section 4, our convergence analysis builds on these propositions, where $R^t$ denotes the residual error at round $t$. Additionally, Corollary 3.5 shows that PERSIA completely mitigates the symmetry induced by LoRA adapters, under which the same full-rank functionality can be realized by distinct symmetric representations. All proofs are provided in Appendix B.

## 4. Convergence and Complexity Analysis

We analyze PERSIA under the standard nonconvex federated optimization framework. Client $i \in [N]$ has objective $f_i(\Theta) := \mathbb{E}_{\xi \sim D_i}[\ell(\Theta; \xi)]$, and the global objective is $f(\Theta) := \frac{1}{N} \sum_{i=1}^{N} f_i(\Theta)$. We view $\Theta$ as the collection of *induced LoRA weight updates* across all adapted layers, and define $\| \Theta \|^2 := \sum_\ell \| \Delta W_\ell \|_F^2$, denoting that the forward model depends on LoRA only through induced updates.

---

**Algorithm 1** PERSIA for FL

1: **Input:** $N$ clients, $R$ rounds, LoRA rank $r$, datasets $\{D_i\}_{i=1}^N$, and the same initialization
2:      $A_i \leftarrow \mathcal{N}(0, \sigma)_{r \times d}$,    $B_i \leftarrow 0_{d \times r}$,
3: **Procedure:**
4: **for** $t$ in $1, \dots, T$ **do**
5:    *Client-Side:* For all clients in parallel $i \in [1, N]$
6:    $A_i, B_i \leftarrow \text{Train}(D_i, B_i, A_i)$
7:    Send $A_i$ and $B_i$ to the server
8:    *Server-Side:*
9:    **for** $i$ in $1, \dots, N$ **do**
10:      $G_i \leftarrow (B_i^\top B_i + \epsilon I_r)^{1/2}$      *# Stabilized Gram*
11:      $Z_i \leftarrow G_i A_i$      *# Whitened Representation*
12:    **end for**
13:    stack $Z_i$ into $\tilde{Z} \leftarrow \{Z_i\}_{i=1}^N \in \mathbb{R}^{Nr \times d}$
14:    $(U, \Sigma, V^\top) \leftarrow \text{TruncatedSVD}(\tilde{Z}, r)$
15:    **for** $i$ in $1, \dots, N$ **do**
16:      $Q_i \leftarrow A_i \times V$
17:      $B_i' \leftarrow B_i \times Q_i$      *# update $B_i$ in FL*
18:    **end for**
19:    $A_i \leftarrow V^\top$      *# update $A_i$ in FL*
20:    $\bar{B} \leftarrow \frac{1}{N} \sum_{i=0}^{N} B_i'$      *# average $B_i$ in FL*
21:    broadcast $\bar{B}$ and $V^\top \in \mathbb{R}^{r \times d}$ as $B_i$ and $A_i$.
22: **end for**

---

**Algorithm 2** PERSIA for PFL

1: **Input:** $N$ clients, $R$ rounds, LoRA rank $r$, datasets $\{D_i\}_{i=1}^N$, and the same initialization
2:      $A_i \leftarrow \mathcal{N}(0, \sigma)_{r \times d}$,    $B_i \leftarrow 0_{d \times r}$,
3: **Procedure:**
4: **for** $t$ in $1, \dots, T$ **do**
5:    *Client-Side:* For all clients in parallel $i \in [1, N]$
6:    $A_i, B_i \leftarrow \text{Train}(D_i, B_i, A_i)$
7:    $G_i \leftarrow (B_i^\top B_i + \epsilon I_r)^{1/2}$      *# Stabilized Gram*
8:    $Z_i \leftarrow G_i A_i$      *# Whitened Representation*
9:    Send $Z_i$ to the server
10:    *Server-Side:*
11:    stack $Z_i$ into $\tilde{Z} \leftarrow \{Z_i\}_{i=1}^N \in \mathbb{R}^{Nr \times d}$
12:    $(U, \Sigma, V^\top) \leftarrow \text{TruncatedSVD}(\tilde{Z}, r)$
13:    broadcast $V \in \mathbb{R}^{d \times r}$
14:    *Client-Side:* For all clients in parallel $i \in [1, N]$
15:    $Q_i \leftarrow A_i \times V$
16:    $B_i \leftarrow B_i \times Q_i$      *# update $B_i$ in PFL*
17:    $A_i \leftarrow V^\top$      *# update $A_i$ in PFL*
18: **end for**

---

At communication round $t$, each client initializes at $\Theta^t$ and runs $\tau$ local SGD steps with stepsize $\eta$, producing $\Theta_i^{t,\tau}$. Define the averaged local model $\Theta_{\text{avg}}^t := \frac{1}{N} \sum_{i=1}^{N} \Theta_i^{t,\tau}$.

PERSIA outputs an aggregated model $\overline{\Theta}^{t+1}$ which may differ from $\Theta_{\text{avg}}^t$ because PERSIA enforces a shared rank-$r$ right subspace in low-rank form. We isolate this effect via the *aggregation perturbation*:

$$\Delta_{\text{agg}}^t := \overline{\Theta}^{t+1} - \Theta_{\text{avg}}^t.$$

Based on Propositions 3.3 and 3.4 we can show that $\Delta_{\text{agg}}^t$ is bounded by $R^t/N$, the proof located in Appendix C.

**Assumption 4.1** (Well-defined Gram). *For each layer and client, the Gram factor is computed as $G_i = (B_i^\top B_i + \varepsilon I_r)^{1/2}$ for some $\varepsilon \geq 0$ so that $G_i$ is positive definite.*

**Assumption 4.2** (Smoothness). *Each $f_i$ is L-smooth: for all $\Theta, \Theta'$, $\|\nabla f_i(\Theta) - \nabla f_i(\Theta')\| \leq L\|\Theta - \Theta'\|$.*

**Assumption 4.3** (Unbiased gradients and bounded variance). *Each client uses stochastic gradients $g_i(\Theta; \xi)$ with $\mathbb{E}[g_i(\Theta; \xi)] = \nabla f_i(\Theta)$ and $\mathbb{E}\|g_i(\Theta; \xi) - \nabla f_i(\Theta)\|^2 \leq \sigma^2$.*

**Assumption 4.4** (Bounded gradient second moment). *There exists $G > 0$ such that for all iterates that appear in the algorithm,*

$$E\|\nabla f_i(\Theta)\|^2 \leq G^2.$$

*In particular, $E\|\nabla f_i(\Theta_i^{t,s})\|^2 \leq G^2$ for all $t, s$.*

**Assumption 4.5** (Bounded gradient second moments). *For every round $t$, client $i$, adapted layer $\ell$, each per-step stochastic gradient has a finite second moment $\mathbb{E}\big[\|\nabla_A \ell_i\|_F^2\big] \leq \sigma_A^2$, $\mathbb{E}\big[\|\nabla_B \ell_i\|_F^2\big] \leq \sigma_B^2$.*

**Assumption 4.6** (Bounded B-iterates). *For every round $t$, client $i$, and adapted layer $\ell$, $\mathbb{E}\big\|B_{i,\ell}^t\big\|_F^2 \leq C_{B,\ell}^2$ .*

**Theorem 4.7** (Convergence PERSIA-FL to Stationary). *Under Assumptions 4.1 to 4.4, and $0 \leq \eta \leq \frac{1}{L}$. Then there exists a constant $M_{FL} > 0$, which depends on the constants specified in the Assumptions, such that*

$$\frac{1}{T} \sum_{t=0}^{T-1} E\|\nabla f(\Theta^t)\|^2 \leq \frac{8}{\tau}\sqrt{\frac{DM_{FL}}{T}} = O\left(\frac{1}{\sqrt{T}}\right). \quad (7)$$

**Theorem 4.8** (Convergence PERSIA-PFL to Stationary). *Under Assumptions 4.1 to 4.4, and $0 \leq \eta \leq \frac{1}{L}$. Then there exists a constant $M_{PFL} > 0$, which depends on the constants specified in the Assumptions, such that*

$$\frac{1}{NT} \sum_{i=1}^{N} \sum_{t=0}^{T-1} E\|\nabla f_i(\Theta_i^t)\|^2 \leq \frac{8}{\tau}\sqrt{\frac{DM_{PFL}}{T}} = O\left(\frac{1}{\sqrt{T}}\right). \quad (8)$$

**Discussion.** Theorems 4.7 and 4.8 shows that PERSIA converges to a stationary point at a rate of $O\left(1/\sqrt{T}\right)$ under smooth non-convex objectives, which matches the optimal convergence rate of FedAvg in the same setting. $M_{FL}$ and $M_{PFL}$ refer to constants that are conditioned with $\sigma^2, G, L, \tau, \sum_{\ell,\sigma_A^2}(C_{B,\ell}^2 + \varepsilon)$ and $\eta$ as described in the Appendix C.

### 4.1. Complexity Analysis

This section summarizes the main computational and memory trade-offs of PERSIA and competing aggregation schemes. All detailed derivations, exact memory, and FLOP counts are deferred to Appendix D.

Table 1 highlights the dominant order of scaling, considering a single communication round and a single LoRA-adapted matrix (layer). Let $N$ denote the number of clients, $d$ the hidden dimension, and $r \ll d$ the LoRA rank. Results scale linearly with the number of layers $L$. Because the aggregation of each layer is independent of the others, all layers can be aggregated in parallel or sequentially, allowing the scale of $L$ to appear in either the time or memory complexity.

The results show that PERSIA trades additional computation for substantially improved aggregation quality, while avoiding the prohibitive $d^2$ memory and time costs of full-rank SVD-based merging. In practice, this trade-off is acceptable because aggregation is performed infrequently relative to local training, and server-side resources are typically less constrained than client-side communication.

*Table 1.* Time and memory complexity per communication round.

| Method | Time complexity | Memory complexity |
|---|---|---|
| Factor-wise | $\mathcal{O}(Ndr)$ | $\mathcal{O}(Ndr)$ |
| Full-rank + SVD | $\mathcal{O}(Nd^2r + d^2r)$ | $\mathcal{O}(2Ndr + d^2)$ |
| PERSIA (ours) | $\mathcal{O}(Ndr^2)$ | $\mathcal{O}(Ndr)$ |

## 5. Experimental Setup

We evaluate the proposed method on natural language understanding using RoBERTa-Large with 355M parameters (Liu et al., 2019) and benchmark it on the GLUE suite (Wang et al., 2018), including MNLI, SST-2, QNLI, QQP, and RTE tasks. All implementations are built on the FederatedScope-LLM (Kuang et al., 2023), and experiments are conducted on NVIDIA Tesla P100 GPUs with 16GB of VRAM and half-precision enabled. Each run uses a single GPU and takes approximately 200 minutes. In total, reproducing results from three independent runs for 60 main experiments and a single run for 500 supplementary experiments (680 runs in total) requires 2267 GPU-hours.

The primary results in Tables 2 and 3 report convergence accuracy over 200 training rounds, with each client taking 40 local steps. All results are averaged over three measurements, with both mean and standard deviation provided.

**Non-IID Data Partitioning.** To simulate realistic federated learning conditions, we distribute the training data across 10 clients using a non-IID partitioning scheme. Client datasets are generated from a Dirichlet distribution with concentration parameter $\alpha = 0.5$ over class labels, resulting in heterogeneous label proportions across clients.

**Federated Learning Protocol.** In each communication round, 30% of clients are sampled uniformly to participate. For global FL, the server evaluates the aggregated model using an independent test dataset. In contrast, for PFL, each client reserves 10% of its local data for evaluation, and the reported accuracy is the average across all clients.

Table 2. Test accuracy (%) on GLUE benchmarks under **Global FL**.

| Method | MNLI | SST-2 | QNLI | QQP | RTE |
|---|---|---|---|---|---|
| Factor-wise | $85.20_{\pm 0.18}$ | $68.32_{\pm 0.01}$ | $87.87_{\pm 0.11}$ | $64.64_{\pm 0.01}$ | $96.65_{\pm 0.15}$ |
| FedProx (Li et al., 2020) | $85.22_{\pm 0.18}$ | $94.45_{\pm 0.12}$ | $89.84_{\pm 0.12}$ | $66.37_{\pm 0.01}$ | $99.53_{\pm 0.15}$ |
| FedMA (Wang et al., 2020) | $85.53_{\pm 0.17}$ | $94.81_{\pm 0.11}$ | $89.95_{\pm 0.07}$ | $65.55_{\pm 0.01}$ | $97.41_{\pm 0.14}$ |
| FFA-LoRA (Sun et al., 2024) | $85.74_{\pm 0.13}$ | $94.83_{\pm 0.09}$ | $79.50_{\pm 0.11}$ | $71.59_{\pm 0.01}$ | $96.85_{\pm 0.15}$ |
| PERSIA (w/o whitening) | $83.34_{\pm 0.18}$ | $94.77_{\pm 0.11}$ | $82.34_{\pm 0.09}$ | $65.10_{\pm 0.01}$ | $97.18_{\pm 0.14}$ |
| **PERSIA** | $\mathbf{86.93}_{\pm 0.52}$ | $\underline{95.18}_{\pm 0.10}$ | $\underline{90.02}_{\pm 0.14}$ | $\underline{75.16}_{\pm 0.04}$ | $\mathbf{99.56}_{\pm 0.16}$ |
| Full-rank (Stoica et al., 2024a) | $\underline{86.62}_{\pm 0.21}$ | $\mathbf{95.20}_{\pm 0.09}$ | $\mathbf{90.12}_{\pm 0.11}$ | $\mathbf{84.67}_{\pm 0.03}$ | $\mathbf{99.56}_{\pm 0.15}$ |

Table 3. Test accuracy (%) on GLUE benchmarks under **Personalized FL**.

| Method | MNLI | SST-2 | QNLI | QQP | RTE |
|---|---|---|---|---|---|
| Factor-wise | $86.37_{\pm 0.23}$ | $60.19_{\pm 0.09}$ | $87.42_{\pm 0.09}$ | $69.15_{\pm 0.07}$ | $87.63_{\pm 0.12}$ |
| FedProx (Li et al., 2020) | $\underline{86.82}_{\pm 0.12}$ | $95.25_{\pm 0.10}$ | $\underline{90.80}_{\pm 0.09}$ | $\underline{73.06}_{\pm 0.07}$ | $98.23_{\pm 0.05}$ |
| FedMA (Wang et al., 2020) | $86.61_{\pm 0.15}$ | $95.02_{\pm 0.12}$ | $87.60_{\pm 0.09}$ | $72.11_{\pm 0.07}$ | $\underline{98.64}_{\pm 0.14}$ |
| PERSIA (w/o whitening) | $80.59_{\pm 0.16}$ | $92.27_{\pm 0.14}$ | $85.49_{\pm 0.12}$ | $70.19_{\pm 0.07}$ | $83.63_{\pm 0.09}$ |
| **PERSIA** | $\mathbf{87.01}_{\pm 0.11}$ | $\mathbf{97.12}_{\pm 0.09}$ | $\mathbf{91.27}_{\pm 0.09}$ | $\mathbf{85.90}_{\pm 0.07}$ | $\mathbf{99.01}_{\pm 0.14}$ |

**Baselines.** We consider factor-wise averaging (McMahan et al., 2016) and full-rank aggregation followed by SVD decomposition (Stoica et al., 2024a) as the main baselines, which serve as a lower and upper bound on aggregation performance. Additionally, we include FedProx (Li et al., 2020), which improves aggregation stability via proximal regularization, and FedMA (Wang et al., 2020), which performs permutation-based alignment prior to layer-wise averaging, and FFA-LoRA (Sun et al., 2024), which mitigates low-rank averaging errors by freezing the $A$ factors, to serve as a diverse group of baseline techniques.

In personalized FL, where the $B$ factors are local, full-rank aggregation, FFA-LoRA, and common LoRA merging methods (Yadav et al., 2023; Ilharco et al., 2022; Putterman et al., 2024; Zhao et al., 2024) are no longer applicable. Therefore, we compare PERSIA only with factor-wise averaging, FedProx, and FedMA. Finally, to isolate the effect of Gram whitening, we report results for PERSIA without whitening as an additional baseline, in which $A_i$ is sent directly to the server rather than $Z_i$, thereby providing an ablation of its contribution.

**Hyperparameters.** Except for the learning rate, all hyperparameters are shared across baselines. We use a token length and a batch size of $128$. We adopt a LoRA adapter with rank $r = 8$, $\alpha = 16$, and a dropout rate of $0.05$. For each task and method, the learning rate is selected from the fixed set $\eta \in \{0.01, 0.02, 0.03, 0.04, 0.05, 0.06\}$ based on the best validation accuracy achieved after 50 communication rounds. The learning rate values for each baseline and task are presented in Appendix A.

**Additional Experiments.** We hypothesize that the use of a conservative number of local steps in existing LoRA-based FL methods is primarily due to aggregation error, and that improved aggregation enables stable training with a larger number of local steps. To evaluate this hypothesis, we conduct additional experiments that vary the number of local epochs from 1 to 640 and compare the performance of all baselines across this range. A summary is shown in Fig. 5, with the full results located in Appendix E.

## 6. Results and Discussion

Tables 2 and 3 reports test accuracy in global and personalized FL. Overall, PERSIA consistently outperforms baselines, and in global FL, it closely matches the full-rank upper bound while remaining fully low-rank. Additionally, Fig. 5 shows that these aggregation improvements translate into longer local training without collapse, thereby improving communication efficiency by requiring fewer FL rounds.

**Global FL.** Table 2 shows that PERSIA is consistently strong across all tasks and is particularly effective in regimes where factor-wise averaging breaks down due to the low-rank averaging mismatch. On SST-2, factor-wise averaging achieves 68.32%, whereas PERSIA achieves 95.18% (a +26.86-point gain). On QQP, factor-wise averaging achieves 64.64% while PERSIA improves to 75.16% (+10.52-points). PERSIA also improves over FedProx and FedMA on QQP, increasing from 66.37% (FedProx) and 65.55% (FedMA) to 75.16%.

Moreover, PERSIA can match the full-rank reference. On MNLI, PERSIA achieves 86.93% versus 86.62% for Full-

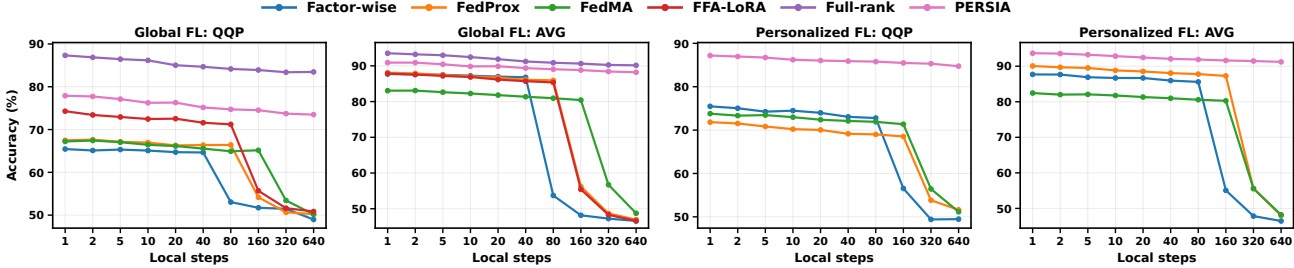

*Figure 5.* Comparison of test accuracy versus the number of local steps at each round, under a fixed number of training batches in total. The results present the QQP task and the average across all tasks for both global and personalized FL.

rank, and on RTE, both achieve 99.56%. On QNLI and SST-2, PERSIA remains within a narrow margin of the full-rank reference (90.02% vs. 90.12% on QNLI, and 95.18% vs. 95.20% on SST-2), providing direct empirical evidence that the proposed low-rank procedure reproduces the effect of full-rank averaging without materializing dense updates.

**Personalized FL.** In PFL, the server cannot access $B_i$, so full-rank aggregation is inapplicable, and the mismatch induced by sharing only one factor becomes more pronounced. Table 3 shows that PERSIA achieves the best performance on all tasks, often with large margins. On SST-2, factor-wise averaging drops to 60.19%, while PERSIA attains 97.12% (a +36.93 point gain). On QQP, PERSIA reaches 85.90%, outperforming FedProx (73.06%) by +12.84 points and factor-wise averaging (69.15%) by +16.75 points. Similar improvements hold across the remaining tasks; for instance, on RTE, PERSIA, it achieves 99.01% versus 98.64% for FedMA and 98.23% for FedProx. These results are consistent with the PFL mechanism of PERSIA: constructing a shared right subspace from whitened representations and enforcing algebraic consistency through rebasing, both of which remain feasible under partial sharing.

**Ablation study.** The ablation in Tables 2 and 3 isolates the effect of Gram whitening. Removing whitening substantially degrades accuracy, confirming that metric correction is required for the server-side SVD to recover a consistent shared right subspace. The effect is especially clear on QNLI, where PERSIA improves from 82.34% (w/o whitening) to 90.02% (a +7.68 point gain), and on MNLI, where it improves from 83.34% to 86.93% (+3.59 points). This directly supports the role of whitening described in Equations (2) and (3), where the right-side representation becomes comparable across clients under a common Euclidean geometry.

**Longer local training.** We vary the local-to-global trade-off by fixing the total local computation and sweeping the number of local steps from 1 to 640, corresponding to a continuum from many communication rounds with short local training to near few-shot ag-

gregation with long local training. Specifically, we chose the constant number of total batches $3200 = \mathcal{E}\mathcal{G}$, where $\mathcal{E} \in \{1, 2, 4, 10, 20, 40, 80, 160, 320, 640\}$ denotes the number of local epochs, and $\mathcal{G}$ denotes the number of federated rounds.

As shown in Fig. 5, factor-wise baselines degrade sharply as local steps increase, whereas PERSIA remains stable across the full range, similarly to the full-rank reference. This stability indicates that reducing low-rank aggregation mismatch enlarges the feasible local training interval, enabling substantial communication reduction without sacrificing convergence. For instance, in global FL, factor-wise aggregation breaks down for $\mathcal{E} > 40$, whereas PERSIA achieves similar performance with $\mathcal{E} = 640$, with $16\times$ higher communication efficiency. Additional results for other tasks are located in Appendix E.

## 7. Conclusion

We studied the low-rank aggregation problem in federated LoRA, in which the factor-wise averaging is inconsistent with the induced updates and becomes ill-posed under client-specific invertible symmetry. We proposed **PERSIA**, which applies Gram-whitening to obtain a metric-correct right-side representation, then uses a truncated SVD to recover a shared rank-$r$ right subspace and rebase client solutions before averaging. This yields a purely low-rank procedure that matches the Frobenius-optimal rank-$r$ projection of the full-rank average, and it naturally extends to personalized FL where only the right factor is shared. Experiments on RoBERTa-Large across GLUE show that PERSIA consistently improves accuracy and stability, remains close to the full-rank upper bound, and enables substantially longer local training without collapse, reducing communication while retaining performance.

An interesting direction for future research is to extend PERSIA to multi-task and multi-agent scenarios. In such settings, LoRA modules are trained independently on distinct tasks or agents and merged into a unified model.

## Impact Statement

This paper presents work whose goal is to advance the field of Machine Learning. There are many potential societal consequences of our work, none of which we feel must be specifically highlighted here.

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

## A. Hyperparameters and Configuration of Baselines

In Table 4, the selected learning rates show a clear shift that depends on the method and task rather than a uniform Global vs. Personalized trend. Factor-wise is consistently more conservative under Personalized FL (mostly 1E-2 to 2E-2) than under Global FL (typically 2E-2 to 3E-2), suggesting that once client-specific heads/solutions are allowed to specialize, smaller steps are sufficient and less noisy. In contrast, stronger parameter-efficient baselines (FFA-LoRA and PERSIA) tend to prefer the upper end of the grid, with Personalized FL often matching or slightly increasing the already large Global-FL choices (e.g., MNLI and QQP at 5E-2). RTE remains stable and conservative across nearly all methods (mostly 2E-2), consistent with its smaller effective supervision, where aggressive updates can destabilize training. Overall, the tuning indicates that personalization does not universally increase the optimal learning rate: it lowers the preferred step size for the simplest baseline (Factor-wise), while leaving high-capacity adapters near their maximal stable rates.

*Table 4.* Selected learning rates for each task and baseline (Global FL vs. Personalized FL).

|  | Setting | MNLI | SST2 | QNLI | QQP | RTE |
|---|---|---|---|---|---|---|
|  | Factor-wise | 2E-2 | 3E-2 | 2E-2 | 2E-2 | 2E-2 |
|  | FFA-LoRA | 4E-2 | 4E-2 | 3E-2 | 4E-2 | 2E-2 |
| Global FL | FedProx | 3E-2 | 4E-2 | 3E-2 | 3E-2 | 2E-2 |
|  | FedMA | 2E-2 | 2E-2 | 3E-2 | 3E-2 | 2E-2 |
|  | Full-rank | 2E-2 | 2E-2 | 2E-2 | 2E-2 | 2E-2 |
|  | PERSIA | 2E-2 | 2E-2 | 2E-2 | 2E-2 | 2E-2 |
|  | Factor-wise | 1E-2 | 2E-2 | 1E-2 | 1E-2 | 1E-2 |
|  | FedProx | 4E-2 | 4E-2 | 3E-2 | 4E-2 | 4E-2 |
| PersonaFL | FedMA | 2E-2 | 3E-2 | 3E-2 | 4E-2 | 2E-2 |
|  | PERSIA | 3E-2 | 3E-2 | 3E-2 | 3E-2 | 3E-2 |

## B. Proofs of Propositions

This appendix provides complete, step-by-step proofs and justifications for all mathematical claims in Section 3 whose proofs are omitted from the main text, including Propositions 3.3 and 3.4 and the identities Equations (2), (3), (5) and (6).

### B.1. Standing notation and assumptions

Throughout this appendix, we use the same notation as in Section 3. For a fixed LoRA-injected linear layer, each client $i \in \{1, \ldots, N\}$ produces a rank-$r$ update

$$\Delta W_i := B_i A_i, \qquad B_i \in \mathbb{R}^{d \times r}, \ A_i \in \mathbb{R}^{r \times k}.$$

**Assumption A.1 (Full column rank of $B_i$).** For each client $i$, the matrix $B_i \in \mathbb{R}^{d \times r}$ has full column rank $r$.

*Justification.* The PERSIA client-side construction in Section 3 defines

$$G_i := (B_i^\top B_i)^{1/2}, \qquad O_i := B_i G_i^{-1}.$$

To make $G_i^{-1}$ well-defined as an ordinary inverse, we need $B_i^\top B_i$ to be invertible. Now $B_i^\top B_i$ is invertible if and only if $\mathrm{rank}(B_i) = r$ (full column rank), because for any $x \in \mathbb{R}^r$,

$$(B_i^\top B_i)x = 0 \iff x^\top (B_i^\top B_i)x = 0 \iff \|B_i x\|_2^2 = 0 \iff B_i x = 0 \iff x = 0 \quad (\text{if } \mathrm{rank}(B_i) = r).$$

Hence $B_i^\top B_i$ is symmetric positive definite, has a (unique) symmetric positive definite square root, and is invertible. This makes $G_i$ and $O_i$ are well-defined in the exact (unregularized) form used in the main text. □

**Assumption A.2 (Stabilized square root when needed).** When $B_i^\top B_i$ is ill-conditioned (Assumption A.1 does not hold), we may compute $G_{i,\varepsilon} := (B_i^\top B_i + \varepsilon I_r)^{1/2}$ for $\varepsilon > 0$.

*Justification.* For any $\varepsilon > 0$ and any nonzero $x \in \mathbb{R}^r$,

$$x^\top(B_i^\top B_i + \varepsilon I_r)x = x^\top(B_i^\top B_i)x + \varepsilon x^\top x = \|B_i x\|_2^2 + \varepsilon\|x\|_2^2 > 0.$$

Therefore $B_i^\top B_i + \varepsilon I_r$ is symmetric positive definite and invertible. Hence $(B_i^\top B_i + \varepsilon I_r)^{1/2}$ exists uniquely as a symmetric positive definite matrix, and its inverse exists as well. This guarantees numerical stability for the same algebraic constructions. $\square$

### B.2. Auxiliary lemmas used by PERSIA

**Lemma B.1** (PERSIA Gram-whitening identities). *Fix a client $i$ and assume $B_i$ has full column rank. Define $G_i, O_i$ as in the main text, and define*

$$Z_i := G_i A_i.$$

*Then the identities in the main text hold:*

$$\|\Delta W_i\|_F^2 = \operatorname{tr}(A_i^\top B_i^\top B_i A_i) = \|G_i A_i\|_F^2 = \|Z_i\|_F^2, \tag{9}$$

$$\|\Delta W_i(I - P)\|_F^2 = \|Z_i(I - P)\|_F^2 \quad \text{for any } P = P^\top = P^2 \in \mathbb{R}^{k \times k}. \tag{10}$$

*In particular, Equations* (9) *and* (10) *are exactly Equations* (2) *and* (3).

*Proof.* Recall $\Delta W_i = B_i A_i$.

**Proof of Equation (9).** Using the Frobenius norm definition,

$$\|\Delta W_i\|_F^2 = \|(B_i A_i)\|_F^2 = \operatorname{tr}\big((B_i A_i)^\top(B_i A_i)\big) = \operatorname{tr}\big(A_i^\top B_i^\top B_i A_i\big).$$

Now use $B_i^\top B_i = G_i^2$ (since $G_i = (B_i^\top B_i)^{1/2}$):

$$\operatorname{tr}\big(A_i^\top B_i^\top B_i A_i\big) = \operatorname{tr}\big(A_i^\top G_i^2 A_i\big).$$

Therefore

$$\operatorname{tr}\big(A_i^\top G_i^2 A_i\big) = \operatorname{tr}\big((G_i A_i)^\top(G_i A_i)\big) = \|G_i A_i\|_F^2$$

Finally, $Z_i := G_i A_i$, so $\|G_i A_i\|_F^2 = \|Z_i\|_F^2$.

**Proof of Equation (10).** Let $P = P^\top = P^2$ be any orthogonal projector . As in the main text

$$\Delta W_i = B_i A_i = O_i G_i A_i = O_i Z_i.$$

Right-multiply by $(I - P)$:

$$\Delta W_i(I - P) = (O_i Z_i)(I - P) = O_i\big(Z_i(I - P)\big).$$

We know $O_i$ has orthogonal columns, so we have:

$$\|\Delta W_i(I - P)\|_F = \|O_i(Z_i(I - P))\|_F = \|Z_i(I - P)\|_F.$$

Squaring both sides yields Equation (10). $\square$

**Lemma B.2** (Stacking identity). *Let $Z_1, \ldots, Z_N$ be matrices with the same number of columns, and define the vertical stack*

$$\tilde{Z} := \begin{bmatrix} Z_1 \\ Z_2 \\ \vdots \\ Z_N \end{bmatrix}.$$

*Then for any matrix $M$ with a compatible number of columns,*

$$\|\tilde{Z}M\|_F^2 = \sum_{i=1}^{N} \|Z_i M\|_F^2.$$

In particular, if $M = I - P$ then

$$\|\tilde{Z}(I - P)\|_F^2 = \sum_{i=1}^{N} \|Z_i(I - P)\|_F^2.$$

*Proof.* By the definition of block matrix multiplication,

$$\tilde{Z}M = \begin{bmatrix} Z_1 M \\ Z_2 M \\ \vdots \\ Z_N M \end{bmatrix}.$$

By the definition of Frobenius norm squared as the sum of squares of all entries, the squared Frobenius norm of a vertical stack equals the sum of the squared Frobenius norms of the blocks:

$$\left\| \begin{bmatrix} Z_1 M \\ \vdots \\ Z_N M \end{bmatrix} \right\|_F^2 = \sum_{i=1}^{N} \|Z_i M\|_F^2.$$

$\square$

### B.3. Proof of Proposition 3.3

**Proposition B.3** (PERSIA error in FL ). *Given $\tilde{Z}$ and $V_r$ as in (4) and (5), and having $A^\star = V_r^\top$, and $B^\star = \frac{1}{N} \sum_{i=1}^{N} B_i(A_i V_r)$. Then*

$$E_{\mathbf{FL}}(A^\star, B^\star)^2 \leq \frac{1}{N} \|\tilde{Z} - \tilde{Z} V_r V_r^\top\|_F^2 = \frac{\mathcal{R}}{N},$$

*where $\mathcal{R} := \sum_{j>r} \sigma_j^2(\tilde{Z}) = \min_{V^\top V = I_r} \|\tilde{Z} - \tilde{Z} V V^\top\|_F^2$.*

*Proof.* We prove the inequality and the identification of the residual with the singular-value tail.

In Proposition 3.3, PERSIA sets

$$A^\star := V_r^\top, \qquad B^\star := \frac{1}{N} \sum_{i=1}^{N} B_i(A_i V_r).$$

Therefore

$$B^\star A^\star = \left( \frac{1}{N} \sum_{i=1}^{N} B_i(A_i V_r) \right) V_r^\top = \frac{1}{N} \sum_{i=1}^{N} B_i(A_i V_r) V_r^\top$$

$$= \frac{1}{N} \sum_{i=1}^{N} B_i A_i (V_r V_r^\top) = \left( \frac{1}{N} \sum_{i=1}^{N} \Delta W_i \right) (V_r V_r^\top).$$

By definition, $\overline{W} := \frac{1}{N} \sum_{i=1}^{N} \Delta W_i$, so

$$B^\star A^\star = \overline{W}(V_r V_r^\top).$$

By Definition 3.1,

$$E_{\mathbf{FL}}(A^\star, B^\star) = \|\overline{W} - B^\star A^\star\|_F = \|\overline{W} - \overline{W}(V_r V_r^\top)\|_F = \|\overline{W}(I - V_r V_r^\top)\|_F.$$

Thus

$$E_{\mathbf{FL}}(A^\star, B^\star)^2 = \left\| \overline{W}(I - V_r V_r^\top) \right\|_F^2.$$

Let $X_i := \Delta W_i(I - V_r V_r^\top)$. Then

$$\overline{W}(I - V_r V_r^\top) = \left( \frac{1}{N} \sum_{i=1}^{N} \Delta W_i \right) (I - V_r V_r^\top) = \frac{1}{N} \sum_{i=1}^{N} X_i.$$

By the triangle inequality and Cauchy–Schwarz, we have

$$\left\| \frac{1}{N} \sum_{i=1}^{N} X_i \right\|_F \le \frac{1}{N} \sum_{i=1}^{N} \|X_i\|_F \le \left( \frac{1}{N} \sum_{i=1}^{N} \|X_i\|_F^2 \right)^{1/2}.$$

Squaring both sides yields the desired bound:

$$\left\| \frac{1}{N} \sum_{i=1}^{N} X_i \right\|_F^2 \le \frac{1}{N} \sum_{i=1}^{N} \|X_i\|_F^2.$$

Recall $X_i = \Delta W_i (I - V_r V_r^\top)$. By Lemma B.1 with $P = V_r V_r^\top$,

$$\|\Delta W_i (I - V_r V_r^\top)\|_F^2 = \|Z_i (I - V_r V_r^\top)\|_F^2.$$

Therefore

$$\frac{1}{N} \sum_{i=1}^{N} \|X_i\|_F^2 = \frac{1}{N} \sum_{i=1}^{N} \|Z_i (I - V_r V_r^\top)\|_F^2.$$

By Lemma B.2 with $M = I - V_r V_r^\top$,

$$\sum_{i=1}^{N} \|Z_i (I - V_r V_r^\top)\|_F^2 = \|\tilde{Z} (I - V_r V_r^\top)\|_F^2 = \|\tilde{Z} - \tilde{Z} V_r V_r^\top\|_F^2.$$

Hence

$$E_{\mathbf{FL}}(A^\star, B^\star)^2 = \left\| \frac{1}{N} \sum_{i=1}^{N} X_i \right\|_F^2 \le \frac{1}{N} \|\tilde{Z} - \tilde{Z} V_r V_r^\top\|_F^2.$$

By the optimality of the SVD solution, since $V_r$ is the optimizer of Equation (5),

$$\|\tilde{Z} - \tilde{Z} V_r V_r^\top\|_F^2 = \min_{V^\top V = I_r} \|\tilde{Z} - \tilde{Z} V V^\top\|_F^2 = \sum_{j>r} \sigma_j^2(\tilde{Z}).$$

Defining $\mathcal{R} := \sum_{j>r} \sigma_j^2(\tilde{Z})$ yields exactly the claimed bound:

$$E_{\mathbf{FL}}(A^\star, B^\star)^2 \le \frac{1}{N} \|\tilde{Z} - \tilde{Z} V_r V_r^\top\|_F^2 = \frac{\mathcal{R}}{N}.$$

$\square$

### B.4. Proof of Proposition 3.4

**Proposition B.4** (PERSIA error in PFL). *Given $\tilde{Z}$ and $V_r$ as in (4) and (5), and having $A^\star = V_r^\top$, and $B_i^\star = B_i(A_i V_r)$. Then*

$$E_{\mathbf{PFL}}(A^\star)^2 = \frac{1}{N} \|\tilde{Z} - \tilde{Z} V_r V_r^\top\|_F^2 = \frac{\mathcal{R}}{N},$$

*where $\mathcal{R} := \sum_{j>r} \sigma_j^2(\tilde{Z}) = \min_{V^\top V = I_r} \|\tilde{Z} - \tilde{Z} V V^\top\|_F^2$.*

*Proof.* In Proposition 3.4, the notation $B^\star = B_i(A_i V_r)$ is client-dependent; in this proof we write it as

$$B_i^\star := B_i(A_i V_r), \qquad A^\star := V_r^\top.$$

By definition, as in the main text,

$$B_i^\star A^\star = \big(B_i(A_i V_r)\big) V_r^\top = \Delta W_i (V_r V_r^\top).$$

Using Definition 3.2 with the client-specific choice $B_i = B_i^\star$ and shared $A' = A^\star$,

$$E_{\mathbf{PFL}}(A^\star)^2 = \frac{1}{N} \sum_{i=1}^{N} \|\Delta W_i - B_i^\star A^\star\|_F^2$$

$$= \frac{1}{N} \sum_{i=1}^{N} \|\Delta W_i - \Delta W_i(V_r V_r^\top)\|_F^2$$

$$= \frac{1}{N} \sum_{i=1}^{N} \|\Delta W_i(I - V_r V_r^\top)\|_F^2.$$

By Lemma B.1 with $P = V_r V_r^\top$,

$$\|\Delta W_i(I - V_r V_r^\top)\|_F^2 = \|Z_i(I - V_r V_r^\top)\|_F^2.$$

Therefore

$$E_{\mathbf{PFL}}(A^\star)^2 = \frac{1}{N} \sum_{i=1}^{N} \|Z_i(I - V_r V_r^\top)\|_F^2.$$

By Lemma B.2 with $M = I - V_r V_r^\top$,

$$\sum_{i=1}^{N} \|Z_i(I - V_r V_r^\top)\|_F^2 = \|\tilde{Z}(I - V_r V_r^\top)\|_F^2 = \|\tilde{Z} - \tilde{Z} V_r V_r^\top\|_F^2.$$

Hence

$$E_{\mathbf{PFL}}(A^\star)^2 = \frac{1}{N} \|\tilde{Z} - \tilde{Z} V_r V_r^\top\|_F^2.$$

By the optimality of the SVD solution,

$$\|\tilde{Z} - \tilde{Z} V_r V_r^\top\|_F^2 = \min_{V^\top V = I_r} \|\tilde{Z} - \tilde{Z} V V^\top\|_F^2 = \sum_{j>r} \sigma_j^2(\tilde{Z}).$$

Defining $\mathcal{R} := \sum_{j>r} \sigma_j^2(\tilde{Z})$ gives

$$E_{\mathbf{PFL}}(A^\star)^2 = \frac{1}{N} \|\tilde{Z} - \tilde{Z} V_r V_r^\top\|_F^2 = \frac{\mathcal{R}}{N},$$

which is exactly the statement of Proposition 3.4. □

### B.5. Proof of Corollary: Mitigate $GL(\cdot)$ Symmetry

**Corollary B.5** (Mitigate $GL(\cdot)$ Symmetry). *Suppose all clients learn the same functionality under various symmetrical representations. $W^* = W_i = BA = (BT_i)(T_i^{-1}A)$, where $T_i \in \mathbb{R}^{r \times r}$ is client-specific invertible transformation. We can show that the PERSIA factorization of $B^*A^*$ is equivalent to the ideal full-rank aggregation.*

$$E_{\mathbf{FL}}(A^*, B^*) := \|\overline{W} - B^*A^*\|_F = 0,$$

$$E_{\mathbf{PFL}}(A^*, B^*) := \left(\frac{1}{N}\sum_{i=1}^{N}\|\Delta W^* - B^*A^*\|_F^2\right)^{1/2} = 0.$$

*Proof.* Since $W_i = (BT_i)(T_i^{-1}A) = BA$ for all $i \in [N]$, the ideal full-rank aggregation satisfies

$$\overline{W} := \frac{1}{N}\sum_{i=1}^{N}W_i = W_i = BA. \tag{11}$$

**Step 1: Metric whitening preserves the exact product.** Define the symmetric factorization for each client as

$$B_i := BT_i \in \mathbb{R}^{d \times r}, \qquad A_i := T_i^{-1}A \in \mathbb{R}^{r \times d}.$$

Assume $B$ has full column rank and each $T_i \in \mathrm{GL}(r)$, then $B_i$ has full column rank and the Gram matrix $B_i^\top B_i \succ 0$. Let

$$G_i := (B_i^\top B_i)^{1/2}, \qquad O_i := B_i G_i^{-1}, \qquad Z_i := G_i A_i.$$

and the whitening step preserves the exact client product:

$$O_i Z_i = (B_i G_i^{-1})(G_i A_i) = B_i A_i = BT_i T_i^{-1}A = BA. \tag{12}$$

**Step 2: The stacked right factors share a common row space.** Let $Z_{\mathrm{stack}} \in \mathbb{R}^{Nr \times d}$ be the vertical stack of $\{Z_i\}_{i=1}^{N}$. From $Z_i = G_i T_i^{-1}A$, define $M_i := G_i T_i^{-1} \in \mathbb{R}^{r \times r}$. Since $G_i \in \mathrm{GL}(r)$ and $T_i \in \mathrm{GL}(r)$, we have $M_i \in \mathrm{GL}(r)$, hence

$$\mathrm{row}(Z_i) = \mathrm{row}(M_i A) = \mathrm{row}(A) \quad \text{for all } i, \qquad \mathrm{row}(Z_{\mathrm{stack}}) = \mathrm{row}(A). \tag{13}$$

Let $\rho := \mathrm{rank}(A) \le r$. Then the rank-$r$ truncated SVD of $Z_{\mathrm{stack}}$ yields right singular vectors $V \in \mathbb{R}^{d \times r}$ whose columns span $\mathrm{row}(Z_{\mathrm{stack}}) = \mathrm{row}(A)$, so

$$A = AVV^\top \quad \Longrightarrow \quad BA = BAVV^\top. \tag{14}$$

**Step 3: PERSIA aggregation recovers the ideal full-rank mean.** PERSIA forms the rebased left factors and using (12)

$$B_i' := B_i A_i V = BAV \in \mathbb{R}^{d \times r},$$

and aggregates

$$B^* := \frac{1}{N}\sum_{i=1}^{N}BAV, \qquad A^* := V^\top.$$

which is independent of $i$, and the reconstructed global model equals

$$B^*A^* = ((BA)V)V^\top = BAVV^\top = BA = \overline{W}, \tag{15}$$

where the second equality uses (14) and the last uses (11). This proves

$$E_{\mathbf{FL}}(A^*, B^*) = \|\overline{W} - B^*A^*\|_F = 0.$$

**Step 4: Personalized error is also zero.** Since $W_i = BA$ for all $i$ and $B^*A^* = BA$, we have $\|W_i - B^*A^*\|_F = 0$ for every $i$, hence

$$E_{\mathbf{PFL}}(A^*, B^*) = \left(\frac{1}{N}\sum_{i=1}^{N}\|W_i - B^*A^*\|_F^2\right)^{1/2} = 0.$$

$\square$

## C. Proof of Theorems

### C.1. Additional definitions and basic inequalities

**Definition C.1** (Induced-update parameter space and norm). *Let $\Theta$ denote the collection of induced LoRA weight updates across adapted layers: $\Theta = \{\Delta W_\ell\}_\ell$. Define the associated Euclidean norm by*

$$\|\Theta\|^2 := \sum_\ell \|\Delta W_\ell\|_F^2.$$

**Definition C.2** (PERSIA aggregation perturbation). *At round $t$, define the averaged local induced-update model*

$$\Theta_{\mathrm{avg}}^t := \frac{1}{N} \sum_{i=1}^N \Theta_i^{t,\tau},$$

*and let PERSIA output $\overline{\Theta}^{t+1}$. Define the aggregation perturbation*

$$\Delta_{\mathrm{agg}}^t := \overline{\Theta}^{t+1} - \Theta_{\mathrm{avg}}^t.$$

**Definition C.3** (Per-round PERSIA principal residual). *For each adapted layer $\ell$ at round $t$, PERSIA forms whitened factors $Z_{i,\ell}^t := G_{i,\ell}^t A_{i,\ell}^{t,\tau}$ and stacks them as $\tilde{Z}_\ell^t \in I\!\!R^{(Nr) \times k_\ell}$. Let $V_{r,\ell}^t$ be the top-$r$ right singular vectors of $\tilde{Z}_\ell^t$. Define the layer residual*

$$\mathcal{R}_\ell^t := \|\tilde{Z}_\ell^t - \tilde{Z}_\ell^t V_{r,\ell}^t (V_{r,\ell}^t)^\top\|_F^2 = \sum_{j>r} \sigma_j^2(\tilde{Z}_\ell^t),$$

*and define the total residual*

$$\mathcal{R}^t := \sum_\ell \mathcal{R}_\ell^t.$$

### C.2. Bounding the PERSIA aggregation perturbation by $\mathcal{R}^t$

**Lemma C.4** (FL aggregation perturbation bound). *In global FL, for each round $t$,*

$$\|\Delta_{\mathrm{agg}}^t\|^2 \leq \frac{\mathcal{R}^t}{N}.$$

*Proof.* Write $\Theta_{\mathrm{avg}}^t = \{\bar{W}_\ell^t\}_\ell$ and $\overline{\Theta}^{t+1} = \{\overline{W}_\ell^{t+1}\}_\ell$. Then by Definition C.1 and Definition C.2,

$$\|\Delta_{\mathrm{agg}}^t\|^2 = \sum_\ell \|\overline{W}_\ell^{t+1} - \bar{W}_\ell^t\|_F^2.$$

Fix a layer $\ell$. PERSIA constructs $\overline{W}_\ell^{t+1}$ by enforcing the shared right factor given by the top-$r$ singular vectors $V_{r,\ell}^t$ of $\tilde{Z}_\ell^t$. By Proposition B.3 applied to this layer at round $t$,

$$\|\overline{W}_\ell^{t+1} - \bar{W}_\ell^t\|_F^2 \leq \frac{1}{N} \|\tilde{Z}_\ell^t - \tilde{Z}_\ell^t V_{r,\ell}^t (V_{r,\ell}^t)^\top\|_F^2 = \frac{1}{N} \mathcal{R}_\ell^t.$$

Sum over $\ell$:

$$\|\Delta_{\mathrm{agg}}^t\|^2 = \sum_\ell \|\overline{W}_\ell^{t+1} - \bar{W}_\ell^t\|_F^2 \leq \sum_\ell \frac{1}{N} \mathcal{R}_\ell^t = \frac{1}{N} \mathcal{R}^t.$$

$\square$

**Lemma C.5** (PFL projection perturbation identity). *In personalized FL, define the post-projection client model $\Theta_i^{t+1}$ as the PERSIA-projected version of $\Theta_i^{t,\tau}$ using the shared right factor at round $t$. Then*

$$\frac{1}{N} \sum_{i=1}^N \|\Theta_i^{t+1} - \Theta_i^{t,\tau}\|^2 = \frac{\mathcal{R}^t}{N}.$$

*Proof.* Fix a layer $\ell$ and a round $t$. By Proposition B.4 applied layerwise at round $t$,

$$\frac{1}{N}\sum_{i=1}^{N}\|\Delta W_{i,\ell}^{t,\tau} - \Delta W_{i,\ell}^{t+1}\|_F^2 = \frac{1}{N}\|\tilde{Z}_\ell^t - \tilde{Z}_\ell^t V_{r,\ell}^t (V_{r,\ell}^t)^\top\|_F^2 = \frac{1}{N}\mathcal{R}_\ell^t.$$

Sum over $\ell$ and use the definition of $\|\cdot\|^2$ from Definition C.1:

$$\frac{1}{N}\sum_{i=1}^{N}\|\Theta_i^{t+1} - \Theta_i^{t,\tau}\|^2 = \sum_\ell \frac{1}{N}\sum_{i=1}^{N}\|\Delta W_{i,\ell}^{t,\tau} - \Delta W_{i,\ell}^{t+1}\|_F^2 = \sum_\ell \frac{1}{N}\mathcal{R}_\ell^t = \frac{1}{N}\mathcal{R}^t.$$

$\square$

## C.3. A sufficient condition ensuring the PERSIA residual term vanishes

**Assumption C.6** (Bounded gradient second moments ). *For every round $t$, client $i$, adapted layer $\ell$, each per-step stochastic gradient has a finite second moment*

$$\mathbb{E}\big[\|\nabla_A \ell_i\|_F^2\big] \leq \sigma_A^2, \qquad \mathbb{E}\big[\|\nabla_B \ell_i\|_F^2\big] \leq \sigma_B^2.$$

**Assumption C.7** (Bounded $B$-iterates ). *For every round $t$, client $i$, and adapted layer $\ell$,*

$$\mathbb{E}\big\|B_{i,\ell}^t\big\|_F^2 \leq C_{B,\ell}^2.$$

**Lemma C.8** (Residual bound from bounded moments and bounded $B$-iterates). *Assume Assumptions 4.1, C.6 and C.7. Assume moreover that, at the start of each round $t$, all clients share the same broadcast right factor $A_\ell^t$ for each adapted layer $\ell$. Then for each round $t$,*

$$\frac{\mathbb{E}[\mathcal{R}^t]}{N} \leq \eta_t^2 \tau^2 \sigma_A^2 \sum_\ell \big(C_{B,\ell}^2 + \varepsilon\big),$$

*where $\varepsilon$ is the Gram stabilization from Assumption 4.1.*

*Proof.* Fix a round $t$ and a layer $\ell$. By definition, $\tilde{Z}_\ell^t$ stacks blocks $Z_{i,\ell}^t = G_{i,\ell}^t A_{i,\ell}^{t,\tau}$.

By definition,

$$\mathcal{R}_\ell^t = \min_{rank(M)\leq r}\|\tilde{Z}_\ell^t - M\|_F^2.$$

Choose the specific rank-$r$ matrix

$$M_\ell^t := \begin{bmatrix} G_{1,\ell}^t A_\ell^t \\ \vdots \\ G_{N,\ell}^t A_\ell^t \end{bmatrix}.$$

Since each block is $G_{i,\ell}^t A_\ell^t$ with the same $A_\ell^t \in \mathbb{R}^{r\times k_\ell}$, the row space of each block is contained in the row space of $A_\ell^t$, so $rank(M_\ell^t) \leq r$. Therefore,

$$\mathcal{R}_\ell^t \leq \|\tilde{Z}_\ell^t - M_\ell^t\|_F^2.$$

By construction,

$$\tilde{Z}_\ell^t - M_\ell^t = \begin{bmatrix} G_{1,\ell}^t(A_{1,\ell}^{t,\tau} - A_\ell^t) \\ \vdots \\ G_{N,\ell}^t(A_{N,\ell}^{t,\tau} - A_\ell^t) \end{bmatrix}.$$

Hence, by Frobenius additivity over vertical stacks,

$$\|\tilde{Z}_\ell^t - M_\ell^t\|_F^2 = \sum_{i=1}^{N}\|G_{i,\ell}^t(A_{i,\ell}^{t,\tau} - A_\ell^t)\|_F^2.$$

For each $i$,

$$\|G_{i,\ell}^t(A_{i,\ell}^{t,\tau} - A_\ell^t)\|_F \le \|G_{i,\ell}^t\|_2 \, \|A_{i,\ell}^{t,\tau} - A_\ell^t\|_F,$$

so

$$\|G_{i,\ell}^t(A_{i,\ell}^{t,\tau} - A_\ell^t)\|_F^2 \le \|G_{i,\ell}^t\|_2^2 \, \|A_{i,\ell}^{t,\tau} - A_\ell^t\|_F^2.$$

By Assumption 4.1, $G_{i,\ell}^t = (B_{i,\ell}^{t\top} B_{i,\ell}^t + \varepsilon I_r)^{1/2}$, so

$$\|G_{i,\ell}^t\|_2^2 = \|B_{i,\ell}^{t\top} B_{i,\ell}^t + \varepsilon I_r\|_2 \le \|B_{i,\ell}^{t\top} B_{i,\ell}^t\|_2 + \varepsilon = \|B_{i,\ell}^t\|_2^2 + \varepsilon \le \|B_{i,\ell}^t\|_F^2 + \varepsilon.$$

Since all clients start from the same broadcast $A_\ell^t$, we have $A_{i,\ell}^{t,0} = A_\ell^t$. Each local step updates $A$ by SGD:

$$A_{i,\ell}^{t,s+1} = A_{i,\ell}^{t,s} - \eta_t \, G_{i,\ell,A}^{t,s},$$

where $G_{i,\ell,A}^{t,s}$ is the stochastic gradient w.r.t. $A$ at local step $s$. Thus

$$A_{i,\ell}^{t,\tau} - A_\ell^t = -\eta_t \sum_{s=0}^{\tau-1} G_{i,\ell,A}^{t,s}.$$

By the inequality $\|\sum_{s=0}^{\tau-1} X_s\|_F^2 \le \tau \sum_{s=0}^{\tau-1} \|X_s\|_F^2$,

$$\|A_{i,\ell}^{t,\tau} - A_\ell^t\|_F^2 \le \eta_t^2 \, \tau \sum_{s=0}^{\tau-1} \|G_{i,\ell,A}^{t,s}\|_F^2.$$

Take expectation and use Assumption C.6:

$$\mathbb{E}\|A_{i,\ell}^{t,\tau} - A_\ell^t\|_F^2 \le \eta_t^2 \, \tau \sum_{s=0}^{\tau-1} \mathbb{E}\|G_{i,\ell,A}^{t,s}\|_F^2 \le \eta_t^2 \, \tau \sum_{s=0}^{\tau-1} \sigma_A^2 = \eta_t^2 \tau^2 \sigma_A^2. \tag{16}$$

From previous results, we have:

$$\mathbb{E}[\mathcal{R}_\ell^t] \le \sum_{i=1}^N \mathbb{E}\Big[(\|B_{i,\ell}^t\|_F^2 + \varepsilon)\|A_{i,\ell}^{t,\tau} - A_\ell^t\|_F^2\Big].$$

Using Assumption C.7 and Equation (16),

$$\mathbb{E}[\mathcal{R}_\ell^t] \le \sum_{i=1}^N (C_{B,\ell}^2 + \varepsilon) \, \eta_t^2 \tau^2 \sigma_A^2 = N(C_{B,\ell}^2 + \varepsilon) \, \eta_t^2 \tau^2 \sigma_A^2.$$

Divide by $N$ and sum over $\ell$:

$$\frac{\mathbb{E}[\mathcal{R}^t]}{N} = \sum_\ell \frac{\mathbb{E}[\mathcal{R}_\ell^t]}{N} \le \eta_t^2 \tau^2 \sigma_A^2 \sum_\ell (C_{B,\ell}^2 + \varepsilon). \tag{17}$$

$$\square$$

## C.4. Technical assumptions needed to make the stationary bounds fully rigorous

**Assumption C.9** (Setup)**.** *Each client $i$ has objective $f_i(\Theta) = E_{\xi \sim D_i}[\ell(\Theta; \xi)]$ and the global objective is $f(\Theta) = \frac{1}{N} \sum_{i=1}^N f_i(\Theta_i)$.*

*At communication round $t$, let $S_t$ be the active set with $|S_t| = N$, and $\Theta_i^t$ be the broadcast model(In the PFL scheme, the parameters $\Theta_i^t$ may vary across clients, whereas in the FL scheme they remain identical). Each active client $i \in S_t$ runs $\tau$ local SGD steps with stepsize $\eta_t$:*

$$\Theta_i^{t,0} = \Theta_i^t, \qquad \Theta_i^{t,s+1} = \Theta_i^{t,s} - \eta_t \, g_i^{t,s}, \quad s = 0, \ldots, \tau - 1,$$

where $g_i^{t,s}$ is a stochastic gradient at $\Theta_i^{t,s}$. Define the post-local model $\Theta_i^{t,\tau}$ and the post-local average

$$\Theta_{\mathrm{avg}}^t = \frac{1}{N} \sum_{i \in S_t} \Theta_i^{t,\tau}.$$

The server computes an orthogonal projector $P^t$:

$$P^t = (P^t)^\top, \qquad (P^t)^2 = P^t,$$

As in Equation (6), in the FL scheme outputs

$$\Theta^{t+1} = \Theta_{\mathrm{avg}}^t P^t.$$

?? and in the PFL scheme outputs

$$\Theta_i^{t+1} = \Theta_i^{t,\tau} P^t.$$

We define the aggregation residual and the client deviation for different schemes

In the FL scheme

$$e_t = \Theta_{\mathrm{avg}}^t - \Theta^{t+1} = \Theta_{\mathrm{avg}}^t (I - P^t), \qquad d_i^t = \Theta_{\mathrm{avg}}^t - \Theta_i^{t,\tau},$$

so that $\Theta^{t+1} - \Theta_i^{t,\tau} = d_i^t - e_t$.

And in the PFL scheme

$$e_i^t = \Theta_i^{t,\tau} - \Theta_i^{t+1} = \Theta_i^{t,\tau} (I - P^t)$$

## C.5. One-round progress lemmas

From Assumption 4.3, expanding the second moment conditionally gives

$$E[\|g_i^{t,s}\|^2 \mid \Theta_i^{t,s}] = \|\nabla f_i(\Theta_i^{t,s})\|^2 + E[\|g_i^{t,s} - \nabla f_i(\Theta_i^{t,s})\|^2 \mid \Theta_i^{t,s}] \le \|\nabla f_i(\Theta_i^{t,s})\|^2 + \sigma^2.$$

Taking expectation and using Assumption 4.4,

$$E\|g_i^{t,s}\|^2 \le G^2 + \sigma^2. \tag{18}$$

By the local update recursion,

$$\Theta_i^{t,\tau} = \Theta^t - \eta_t \sum_{s=0}^{\tau-1} g_i^{t,s}, \qquad \Theta_{\mathrm{avg}}^t = \Theta^t - \eta_t \sum_{s=0}^{\tau-1} \bar{g}^{t,s}, \quad \bar{g}^{t,s} := \frac{1}{N} \sum_{j \in S_t} g_j^{t,s}.$$

Hence the deviation is

$$d_i^t = \Theta_{\mathrm{avg}}^t - \Theta_i^{t,\tau} = -\eta_t \sum_{s=0}^{\tau-1} (\bar{g}^{t,s} - g_i^{t,s}) = \eta_t \sum_{s=0}^{\tau-1} (g_i^{t,s} - \bar{g}^{t,s}).$$

Using $\|\sum_{s=0}^{\tau-1} u_s\|^2 \le \tau \sum_{s=0}^{\tau-1} \|u_s\|^2$,

$$\|d_i^t\|^2 \le \eta_t^2 \tau \sum_{s=0}^{\tau-1} \|g_i^{t,s} - \bar{g}^{t,s}\|^2.$$

Also $\|a - b\|^2 \le 2\|a\|^2 + 2\|b\|^2$ and Jensen imply $\|\bar{g}^{t,s}\|^2 \le \frac{1}{N} \sum_{j \in S_t} \|g_j^{t,s}\|^2$. Therefore,

$$E\|g_i^{t,s} - \bar{g}^{t,s}\|^2 \le 2E\|g_i^{t,s}\|^2 + 2E\|\bar{g}^{t,s}\|^2 \le 2(G^2 + \sigma^2) + 2(G^2 + \sigma^2) = 4(G^2 + \sigma^2),$$

using (18). Summing over $s$ yields

$$E\|d_i^t\|^2 \le 4(G^2 + \sigma^2) \eta_t^2 \tau^2. \tag{19}$$

By the definition,

$$e_t = \Theta_{\mathrm{avg}}^t - \Theta^{t+1} = \Theta_{\mathrm{avg}}^t (I - P^t), \qquad \|e_t\|^2 = \|\Theta_{\mathrm{avg}}^t (I - P^t)\|^2.$$

*Under the PERSIA server step, $P^t$ is the optimal rank-$r$ right projector computed from the stacked whitened matrices; denote the corresponding stacked tail residual by $R^t$. Then*

$$E\|e_t\|^2 \leq \frac{E[R^t]}{N}. \tag{20}$$

*As in the PERSIA residual analysis (exact SVD), Equation (17) yields*

$$\frac{E[R^t]}{N} \leq \eta_t^2 \tau^2 \, \sigma_A^2 \sum_\ell (C_{B,\ell}^2 + \varepsilon). \tag{21}$$

*Combining (20) and (21) gives*

$$E\|e_t\|^2 \leq \eta_t^2 \tau^2 \, \sigma_A^2 \sum_\ell (C_{B,\ell}^2 + \varepsilon). \tag{22}$$

**Lemma C.11** (Relating local-path gradients to the initial gradient). *Fix a client $i$ and a communication round $t$. Let the local iterates $\{\Theta_i^{t,s}\}_{s=0}^{\tau-1}$ be generated by $\tau$ steps of local SGD starting from $\Theta_i^{t,0} = \Theta_i^t$ with stepsize $\eta_t$. Then*

$$\frac{\eta_t}{2} \sum_{s=0}^{\tau-1} E\|\nabla f_i(\Theta_i^{t,s})\|^2 \; \geq \; \frac{\eta_t \tau}{4} \, E\|\nabla f_i(\Theta_i^t)\|^2 \; - \; \frac{L^2}{2} \, (G^2 + \sigma^2) \, \eta_t^3 \, \tau^3.$$

*Equivalently,*

$$\frac{\eta_t \tau}{4} \, E\|\nabla f_i(\Theta_i^t)\|^2 \; \leq \; \frac{\eta_t}{2} \sum_{s=0}^{\tau-1} E\|\nabla f_i(\Theta_i^{t,s})\|^2 \; + \; \frac{L^2}{2} \, (G^2 + \sigma^2) \, \eta_t^3 \, \tau^3.$$

*Proof.* By $L$-smoothness of $f_i$, the gradient is $L$-Lipschitz:

$$\|\nabla f_i(U) - \nabla f_i(V)\| \leq L\|U - V\| \quad \forall U, V.$$

Thus for each local step index $s$,

$$\|\nabla f_i(\Theta_i^{t,s})\| \geq \|\nabla f_i(\Theta_i^t)\| - \|\nabla f_i(\Theta_i^{t,s}) - \nabla f_i(\Theta_i^t)\| \geq \|\nabla f_i(\Theta_i^t)\| - L\|\Theta_i^{t,s} - \Theta_i^t\|.$$

Using the inequality $(a-b)^2 \geq \frac{1}{2}a^2 - b^2$ for all real $a, b$, we obtain

$$\|\nabla f_i(\Theta_i^{t,s})\|^2 \geq \frac{1}{2}\|\nabla f_i(\Theta_i^t)\|^2 - L^2\|\Theta_i^{t,s} - \Theta_i^t\|^2.$$

Taking expectation and summing $s = 0, \ldots, \tau - 1$ gives

$$\sum_{s=0}^{\tau-1} E\|\nabla f_i(\Theta_i^{t,s})\|^2 \geq \frac{\tau}{2} E\|\nabla f_i(\Theta_i^t)\|^2 - L^2 \sum_{s=0}^{\tau-1} E\|\Theta_i^{t,s} - \Theta_i^t\|^2. \tag{23}$$

It remains to upper bound $\sum_{s=0}^{\tau-1} E\|\Theta_i^{t,s} - \Theta_i^t\|^2$.

From the local SGD recursion,

$$\Theta_i^{t,s} - \Theta_i^t = -\eta_t \sum_{k=0}^{s-1} g_i^{t,k}.$$

By Cauchy–Schwarz, $\|\sum_{k=0}^{s-1} u_k\|^2 \leq s \sum_{k=0}^{s-1} \|u_k\|^2$, hence

$$\|\Theta_i^{t,s} - \Theta_i^t\|^2 \leq \eta_t^2 \, s \sum_{k=0}^{s-1} \|g_i^{t,k}\|^2.$$

Taking expectation and using

$$E\|g_i^{t,k}\|^2 = E\|\nabla f_i(\Theta_i^{t,k}) + (g_i^{t,k} - \nabla f_i(\Theta_i^{t,k}))\|^2 = E\|\nabla f_i(\Theta_i^{t,k})\|^2 + E\|g_i^{t,k} - \nabla f_i(\Theta_i^{t,k})\|^2 \leq G^2 + \sigma^2,$$

We obtain

$$E\|\Theta_i^{t,s} - \Theta_i^t\|^2 \leq \eta_t^2\, s \sum_{k=0}^{s-1}(G^2+\sigma^2) = \eta_t^2\, s^2\, (G^2+\sigma^2).$$

Therefore,

$$\sum_{s=0}^{\tau-1} E\|\Theta_i^{t,s} - \Theta_i^t\|^2 \leq \eta_t^2\, (G^2+\sigma^2) \sum_{s=0}^{\tau-1} s^2 \leq \eta_t^2\, (G^2+\sigma^2)\, \tau^3,$$

since $\sum_{s=0}^{\tau-1} s^2 \leq \tau^3$.

Substituting this bound into (23) yields

$$\sum_{s=0}^{\tau-1} E\|\nabla f_i(\Theta_i^{t,s})\|^2 \geq \frac{\tau}{2} E\|\nabla f_i(\Theta_i^t)\|^2 - L^2\, \eta_t^2\, (G^2+\sigma^2)\, \tau^3.$$

Multiplying both sides by $\eta_t/2$ and rearranging yields

$$\frac{L^2}{2}\,(G^2+\sigma^2)\,\eta_t^3\,\tau^3 + \frac{\eta_t}{2} \sum_{s=0}^{\tau-1} E\|\nabla f_i(\Theta_i^{t,s})\|^2 \geq \frac{\eta_t\tau}{4} E\|\nabla f_i(\Theta_i^t)\|^2,$$

$\square$

**Lemma C.12** (One-round progress for PERSIA-FL). *Assume Assumptions 4.2 to 4.4. Then*

$$\frac{\eta_t\tau}{4}\, E\big\|\nabla f_i(\Theta_i^t)\big\|^2 \leq f_i(\Theta^t) - E[f_i(\Theta^{t+1})]$$

$$+ G\,\eta_t\tau \sqrt{8(G^2+\sigma^2) + 2\sigma_A^2 \sum_\ell (C_{B,\ell}^2 + \varepsilon)}$$

$$+ \frac{L\eta_t^2\tau}{2}\sigma^2 + \eta_t^2\tau^2\Big(4L(G^2+\sigma^2) + L\sigma_A^2 \sum_\ell(C_{B,\ell}^2+\varepsilon)\Big)$$

$$+ \frac{L^2}{2}\,(G^2+\sigma^2)\,\eta_t^3\,\tau^3. \tag{24}$$

*Proof.* **Step 1**: local SGD descent:

Apply Smoothness with $U = \Theta_i^{t,s+1}$ and $V = \Theta_i^{t,s}$:

$$f_i(\Theta_i^{t,s+1}) \leq f_i(\Theta_i^{t,s}) + \langle \nabla f_i(\Theta_i^{t,s}), \Theta_i^{t,s+1} - \Theta_i^{t,s}\rangle + \frac{L}{2}\|\Theta_i^{t,s+1} - \Theta_i^{t,s}\|^2$$

$$= f_i(\Theta_i^{t,s}) - \eta_t\langle\nabla f_i(\Theta_i^{t,s}), g_i^{t,s}\rangle + \frac{L\eta_t^2}{2}\|g_i^{t,s}\|^2.$$

Taking conditional expectation given $\Theta_i^{t,s}$ and using Assumption 4.4 ,

$$E[f_i(\Theta_i^{t,s+1}) \mid \Theta_i^{t,s}] \leq f_i(\Theta_i^{t,s}) - \eta_t\|\nabla f_i(\Theta_i^{t,s})\|^2 + \frac{L\eta_t^2}{2}\big(\|\nabla f_i(\Theta_i^{t,s})\|^2 + \sigma^2\big).$$

Assume $\eta_t \leq 1/L$ so that $\eta_t(1 - \frac{L\eta_t}{2}) \geq \eta_t/2$. Taking full expectation,

$$E[f_i(\Theta_i^{t,s+1})] \leq E[f_i(\Theta_i^{t,s})] - \frac{\eta_t}{2}E\|\nabla f_i(\Theta_i^{t,s})\|^2 + \frac{L\eta_t^2}{2}\sigma^2.$$

Summing over $s = 0, \ldots, \tau-1$ yields

$$E[f_i(\Theta_i^{t,\tau})] \leq f_i(\Theta^t) - \frac{\eta_t}{2} \sum_{s=0}^{\tau-1} E\|\nabla f_i(\Theta_i^{t,s})\|^2 + \frac{L\eta_t^2\tau}{2}\sigma^2. \tag{25}$$

**Step 2**: PERSIA step. Apply smoothness with $U = \Theta^{t+1}$ and $V = \Theta_i^{t,\tau}$:

$$f_i(\Theta^{t+1}) \le f_i(\Theta_i^{t,\tau}) + \langle \nabla f_i(\Theta_i^{t,\tau}), \Theta^{t+1} - \Theta_i^{t,\tau} \rangle + \frac{L}{2} \|\Theta^{t+1} - \Theta_i^{t,\tau}\|^2.$$

Under the (C.9) assumption, $\Theta^{t+1} - \Theta_i^{t,\tau} = d_i^t - e_t$, hence

$$f_i(\Theta^{t+1}) \le f_i(\Theta_i^{t,\tau}) + \langle \nabla f_i(\Theta_i^{t,\tau}), d_i^t - e_t \rangle + \frac{L}{2} \|d_i^t - e_t\|^2. \tag{26}$$

By Cauchy–Schwarz,

$$|\langle \nabla f_i(\Theta_i^{t,\tau}), d_i^t - e_t \rangle| \le \|\nabla f_i(\Theta_i^{t,\tau})\| \, \|d_i^t - e_t\|.$$

Taking expectation and applying Cauchy–Schwarz again, then using Assumption 4.4,

$$E|\langle \nabla f_i(\Theta_i^{t,\tau}), d_i^t - e_t \rangle| \le \sqrt{E\|\nabla f_i(\Theta_i^{t,\tau})\|^2} \, \sqrt{E\|d_i^t - e_t\|^2} \le G\sqrt{E\|d_i^t - e_t\|^2}.$$

Also, $\|a - b\|^2 \le 2\|a\|^2 + 2\|b\|^2$ implies

$$E\|d_i^t - e_t\|^2 \le 2E\|d_i^t\|^2 + 2E\|e_t\|^2.$$

Using (19) and (22),

$$E\|d_i^t - e_t\|^2 \le 2 \cdot 4(G^2 + \sigma^2)\eta_t^2 \tau^2 + 2\eta_t^2 \tau^2 \sigma_A^2 \sum_\ell (C_{B,\ell}^2 + \varepsilon).$$

Therefore,

$$E|\langle \nabla f_i(\Theta_i^{t,\tau}), d_i^t - e_t \rangle| \le G \eta_t \tau \sqrt{8(G^2 + \sigma^2) + 2\sigma_A^2 \sum_\ell (C_{B,\ell}^2 + \varepsilon)}. \tag{27}$$

Similarly,

$$\frac{L}{2} E\|d_i^t - e_t\|^2 \le \frac{L}{2}\left(8(G^2 + \sigma^2)\eta_t^2 \tau^2 + 2\eta_t^2 \tau^2 \sigma_A^2 \sum_\ell (C_{B,\ell}^2 + \varepsilon)\right)$$

$$= 4L(G^2 + \sigma^2)\eta_t^2 \tau^2 + L\eta_t^2 \tau^2 \sigma_A^2 \sum_\ell (C_{B,\ell}^2 + \varepsilon). \tag{28}$$

Taking expectation in (26) and applying (27)–(28) yields

$$E[f_i(\Theta^{t+1})] \le E[f_i(\Theta_i^{t,\tau})] + G \eta_t \tau \sqrt{8(G^2 + \sigma^2) + 2\sigma_A^2 \sum_\ell (C_{B,\ell}^2 + \varepsilon)} + \eta_t^2 \tau^2 \left(4L(G^2 + \sigma^2) + L\sigma_A^2 \sum_\ell (C_{B,\ell}^2 + \varepsilon)\right). \tag{29}$$

Combine Equation (25) and (29):

$$E[f_i(\Theta^{t+1})] \le f_i(\Theta^t) - \frac{\eta_t}{2} \sum_{s=0}^{\tau-1} E\|\nabla f_i(\Theta_i^{t,s})\|^2 + \frac{L\eta_t^2 \tau}{2}\sigma^2$$

$$+ G \eta_t \tau \sqrt{8(G^2 + \sigma^2) + 2\sigma_A^2 \sum_\ell (C_{B,\ell}^2 + \varepsilon)}$$

$$+ \eta_t^2 \tau^2 \left(4L(G^2 + \sigma^2) + L\sigma_A^2 \sum_\ell (C_{B,\ell}^2 + \varepsilon)\right). \tag{30}$$

Rearranging gives

$$\frac{\eta_t}{2} \sum_{s=0}^{\tau-1} E\|\nabla f_i(\Theta_i^{t,s})\|^2 \le f_i(\Theta^t) - E[f_i(\Theta^{t+1})]$$

$$+ G \eta_t \tau \sqrt{8(G^2 + \sigma^2) + 2\sigma_A^2 \sum_\ell (C_{B,\ell}^2 + \varepsilon)}$$

$$+ \frac{L\eta_t^2 \tau}{2}\sigma^2 + \eta_t^2 \tau^2 \left(4L(G^2 + \sigma^2) + L\sigma_A^2 \sum_\ell (C_{B,\ell}^2 + \varepsilon)\right). \tag{31}$$

Then using the Lemma C.11 :

$$\frac{\eta_t \tau}{4} E\|\nabla f_i(\Theta_i^t)\|^2 \leq f_i(\Theta^t) - E[f_i(\Theta^{t+1})]$$

$$+ G \eta_t \tau \sqrt{8(G^2 + \sigma^2) + 2\sigma_A^2 \sum_\ell (C_{B,\ell}^2 + \varepsilon)}$$

$$+ \frac{L\eta_t^2 \tau}{2}\sigma^2 + \eta_t^2 \tau^2 \Big( 4L(G^2 + \sigma^2) + L\sigma_A^2 \sum_\ell (C_{B,\ell}^2 + \varepsilon) \Big)$$

$$+ \frac{L^2}{2}(G^2 + \sigma^2)\,\eta_t^3\,\tau^3. \tag{32}$$

$\square$

### C.6. Proof of Theorem 4.7

**Theorem C.13** (Convergence PERSIA-FL to stationary ). *Assume Assumptions 4.2 to 4.4 and constant stepsize $\eta_t \equiv \eta$. Let $D$ satisfy $f_i(\Theta^0) - f_i^\star \leq D$ for all $i$. Then there exists a constant $M > 0$ such that*

$$\frac{1}{T}\sum_{t=0}^{T-1} E\|\nabla f(\Theta^t)\|^2 \leq \frac{8}{\tau}\sqrt{\frac{DM}{T}} = O\Big(\frac{1}{\sqrt{T}}\Big). \tag{33}$$

*Proof.* Assume a constant stepsize $\eta_t \equiv \eta$ and choose any $M$ that

$$G\tau \sqrt{2\sigma_A^2 \sum_\ell (C_{B,\ell}^2 + \varepsilon) + 8(G^2 + \sigma^2)}\,\eta$$

$$+ \frac{L^2}{2}(G^2 + \sigma^2)\eta_t^3\tau^3 + \left[ \frac{L\tau}{2}\sigma^2 + \tau^2\Big( L\sigma_A^2 \sum_\ell (C_{B,\ell}^2 + \varepsilon) + 4L(G^2 + \sigma^2) \Big) \right]\eta^2 \; \leq \; M\eta^2. \tag{34}$$

Then Lemma C.12 implies

$$E\|\nabla f_i(\Theta_i^t)\|^2 \leq \frac{4}{\eta\tau}\big(f_i(\Theta^t) - E[f_i(\Theta^{t+1})]\big) + \frac{4M\eta}{\tau}. \tag{35}$$

Summing (35) over $t = 0, \ldots, T-1$ telescopes:

$$\sum_{t=0}^{T-1} E\|\nabla f_i(\Theta_i^t)\|^2 \leq \frac{4}{\eta\tau}\big(f_i(\Theta^0) - E[f_i(\Theta^T)]\big) + \frac{4M\eta}{\tau}\,T.$$

Using $f_i(\Theta^0) - E[f_i(\Theta^T)] \leq f_i(\Theta^0) - f_i^\star \leq D$ yields

$$\frac{1}{T}\sum_{t=0}^{T-1} E\|\nabla f_i(\Theta_i^t)\|^2 \leq \frac{4D}{\eta\tau T} + \frac{4M\eta}{\tau}. \tag{36}$$

Choosing $\eta = \sqrt{\frac{D}{MT}}$ gives

$$\frac{1}{T}\sum_{t=0}^{T-1} E\|\nabla f_i(\Theta_i^t)\|^2 \leq \frac{8}{\tau}\sqrt{\frac{DM}{T}}.$$

Thus we can obtain:

$$\frac{1}{NT}\sum_{i=1}^{N}\sum_{t=0}^{T-1} E\|\nabla f_i(\Theta_i^t)\|^2 \leq \frac{8}{\tau}\sqrt{\frac{DM}{T}},$$

By Jensen, $\big\|\frac{1}{N}\sum_{i=1}^{N}\nabla f_i(\Theta_i^t)\big\|^2 \leq \frac{1}{N}\sum_{i=1}^{N}\|\nabla f_i(\Theta_i^t)\|^2$ and $\nabla f(\Theta^t) = \frac{1}{N}\sum_{i=1}^{N}\nabla f_i(\Theta_i^t)$.

Taking expectation and averaging over $t = 0, \ldots, T-1$,

$$\frac{1}{T} \sum_{t=0}^{T-1} E\|\nabla f(\Theta^t)\|^2 \leq \frac{1}{T} \sum_{t=0}^{T-1} \frac{1}{N} \sum_{i=1}^{N} E\|\nabla f_i(\Theta_i^t)\|^2 = \frac{1}{NT} \sum_{i=1}^{N} \sum_{t=0}^{T-1} E\|\nabla f_i(\Theta_i^t)\|^2.$$

Hence,

$$\frac{1}{T} \sum_{t=0}^{T-1} E\|\nabla f(\Theta^t)\|^2 \leq \frac{8}{\tau} \sqrt{\frac{DM}{T}} = O\left(\frac{1}{\sqrt{T}}\right). \tag{37}$$

which is (33). □

**Lemma C.14** (One-round progress for PERSIA-PFL). *Assume Assumptions 4.2 to 4.4. Then*

$$\frac{\eta_t \tau}{4} E\|\nabla f_i(\Theta_i^t)\|^2 \leq E[f_i(\Theta_i^t)] - E[f_i(\Theta_i^{t+1})] + \frac{L\eta_t^2 \tau}{2} \sigma^2$$
$$+ G\sqrt{E\|e_i^t\|^2} + \frac{L}{2} E\|e_i^t\|^2 + \frac{L^2}{2}(G^2 + \sigma^2)\eta_t^3 \tau^3. \tag{38}$$

*Proof.* **Step 1**: local SGD descent. This step is identical to the FL case. Applying Smoothness with $U = \Theta_i^{t,s+1}$ and $V = \Theta_i^{t,s}$, using $\Theta_i^{t,s+1} - \Theta_i^{t,s} = -\eta_t g_i^{t,s}$, then taking conditional expectation and using Assumption 4.3, and finally using $\eta_t \leq 1/L$, yields (exactly as in (25)):

$$E[f_i(\Theta_i^{t,\tau})] \leq E[f_i(\Theta_i^t)] - \frac{\eta_t}{2} \sum_{s=0}^{\tau-1} E\|\nabla f_i(\Theta_i^{t,s})\|^2 + \frac{L\eta_t^2 \tau}{2} \sigma^2. \tag{39}$$

**Step 2**: PERSIA step (PFL). Apply Smoothness with $U = \Theta_i^{t+1}$ and $V = \Theta_i^{t,\tau}$:

$$f_i(\Theta_i^{t+1}) \leq f_i(\Theta_i^{t,\tau}) + \langle \nabla f_i(\Theta_i^{t,\tau}), \Theta_i^{t+1} - \Theta_i^{t,\tau} \rangle + \frac{L}{2}\|\Theta_i^{t+1} - \Theta_i^{t,\tau}\|^2$$
$$= f_i(\Theta_i^{t,\tau}) - \langle \nabla f_i(\Theta_i^{t,\tau}), e_i^t \rangle + \frac{L}{2}\|e_i^t\|^2. \tag{40}$$

Taking expectation and bounding the inner product by magnitude,

$$-E\langle \nabla f_i(\Theta_i^{t,\tau}), e_i^t \rangle \leq E|\langle \nabla f_i(\Theta_i^{t,\tau}), e_i^t \rangle| \leq \sqrt{E\|\nabla f_i(\Theta_i^{t,\tau})\|^2} \sqrt{E\|e_i^t\|^2} \leq G\sqrt{E\|e_i^t\|^2}, \tag{41}$$

where we used Cauchy–Schwarz and Assumption 4.4. Substituting (41) into the expectation of (40) gives

$$E[f_i(\Theta_i^{t+1})] \leq E[f_i(\Theta_i^{t,\tau})] + G\sqrt{E\|e_i^t\|^2} + \frac{L}{2} E\|e_i^t\|^2. \tag{42}$$

Plug (39) into (42) and rearrange:

$$\frac{\eta_t}{2} \sum_{s=0}^{\tau-1} E\|\nabla f_i(\Theta_i^{t,s})\|^2 \leq E[f_i(\Theta_i^t)] - E[f_i(\Theta_i^{t+1})] + \frac{L\eta_t^2 \tau}{2} \sigma^2 + G\sqrt{E\|e_i^t\|^2} + \frac{L}{2} E\|e_i^t\|^2. \tag{43}$$

By Lemma C.11,

$$\frac{\eta_t}{2} \sum_{s=0}^{\tau-1} E\|\nabla f_i(\Theta_i^{t,s})\|^2 \geq \frac{\eta_t \tau}{4} E\|\nabla f_i(\Theta_i^t)\|^2 - \frac{L^2}{2}(G^2 + \sigma^2)\eta_t^3 \tau^3.$$

Combine this lower bound with (43) to obtain

$$\frac{\eta_t \tau}{4} E\|\nabla f_i(\Theta_i^t)\|^2 \leq E[f_i(\Theta_i^t)] - E[f_i(\Theta_i^{t+1})] + \frac{L\eta_t^2 \tau}{2} \sigma^2$$
$$+ G\sqrt{E\|e_i^t\|^2} + \frac{L}{2} E\|e_i^t\|^2 + \frac{L^2}{2}(G^2 + \sigma^2)\eta_t^3 \tau^3. \tag{44}$$

□

## C.7. Proof of Theorem 4.8

**Theorem C.15** (Convergence PERSIA-PFL to stationary). *Assume Assumptions 4.2 to 4.4 and constant stepsize $\eta_t \equiv \eta$. Let $D$ satisfy $f_i(\Theta^0) - f_i^\star \leq D$ for all $i$. Then there exists a constant $M > 0$ such that*

$$\frac{1}{NT} \sum_{i=1}^{N} \sum_{t=0}^{T-1} E\|\nabla f_i(\Theta_i^t)\|^2 \leq \frac{8}{\tau} \sqrt{\frac{DM}{T}} = O\left(\frac{1}{\sqrt{T}}\right). \tag{45}$$

*Proof.* Average (44) over $i \in S_t$:

$$\frac{\eta\tau}{4} \cdot \frac{1}{N} \sum_{i \in S_t} E\|\nabla f_i(\Theta_i^t)\|^2 \leq \frac{1}{N} \sum_{i \in S_t} \left( E[f_i(\Theta_i^t)] - E[f_i(\Theta_i^{t+1})] \right) + \frac{L\eta^2\tau}{2}\sigma^2$$
$$+ \frac{G}{N} \sum_{i \in S_t} \sqrt{E\|e_i^t\|^2} + \frac{L}{2} \cdot \frac{1}{N} \sum_{i \in S_t} E\|e_i^t\|^2 + \frac{L^2}{2}(G^2 + \sigma^2)\eta^3\tau^3. \tag{46}$$

By Jensen, $\frac{1}{N} \sum_{i \in S_t} \sqrt{x_i} \leq \sqrt{\frac{1}{N} \sum_{i \in S_t} x_i}$, hence

$$\frac{G}{N} \sum_{i \in S_t} \sqrt{E\|e_i^t\|^2} \leq G \sqrt{\frac{1}{N} \sum_{i \in S_t} E\|e_i^t\|^2}.$$

Using the perturbation identity Lemma C.5 gives

$$G\sqrt{\frac{1}{N} \sum_{i \in S_t} E\|e_i^t\|^2} = G\sqrt{\frac{E[R^t]}{N}}, \qquad \frac{1}{N} \sum_{i \in S_t} E\|e_i^t\|^2 = \frac{E[R^t]}{N}.$$

Substituting into (46) yields

$$\frac{\eta\tau}{4} \cdot \frac{1}{N} \sum_{i \in S_t} E\|\nabla f_i(\Theta_i^t)\|^2 \leq \frac{1}{N} \sum_{i \in S_t} \left( E[f_i(\Theta_i^t)] - E[f_i(\Theta_i^{t+1})] \right) + \frac{L\eta^2\tau}{2}\sigma^2$$
$$+ G\sqrt{\frac{E[R^t]}{N}} + \frac{L}{2} \cdot \frac{E[R^t]}{N} + \frac{L^2}{2}(G^2 + \sigma^2)\eta^3\tau^3. \tag{47}$$

Using Equation (17),

$$G\sqrt{\frac{E[R^t]}{N}} \leq G\eta\tau\sqrt{\sigma_A^2 \sum_\ell (C_{B,\ell}^2 + \varepsilon)}, \qquad \frac{L}{2} \cdot \frac{E[R^t]}{N} \leq \frac{L}{2}\eta^2\tau^2 \sigma_A^2 \sum_\ell (C_{B,\ell}^2 + \varepsilon).$$

Hence (47) implies

$$\frac{\eta\tau}{4} \cdot \frac{1}{N} \sum_{i \in S_t} E\|\nabla f_i(\Theta_i^t)\|^2 \leq \frac{1}{N} \sum_{i \in S_t} \left( E[f_i(\Theta_i^t)] - E[f_i(\Theta_i^{t+1})] \right)$$
$$+ G\eta\tau\sqrt{\sigma_A^2 \sum_\ell (C_{B,\ell}^2 + \varepsilon)} + \frac{L\eta^2\tau}{2}\sigma^2 + \frac{L}{2}\eta^2\tau^2 \sigma_A^2 \sum_\ell (C_{B,\ell}^2 + \varepsilon) + \frac{L^2}{2}(G^2 + \sigma^2)\eta^3\tau^3. \tag{48}$$

Choose any $M$ such that

$$G\tau\sqrt{\sigma_A^2 \sum_\ell (C_{B,\ell}^2 + \varepsilon)}\,\eta + \left[\frac{L\tau}{2}\sigma^2 + \frac{L}{2}\tau^2 \sigma_A^2 \sum_\ell (C_{B,\ell}^2 + \varepsilon)\right]\eta^2 + \frac{L^2}{2}(G^2 + \sigma^2)\tau^3\,\eta^3 \leq M\eta^2. \tag{49}$$

Then (48) yields

$$\frac{1}{N} \sum_{i \in S_t} E\|\nabla f_i(\Theta_i^t)\|^2 \leq \frac{4}{\eta\tau} \cdot \frac{1}{N} \sum_{i \in S_t} \left( E[f_i(\Theta_i^t)] - E[f_i(\Theta_i^{t+1})] \right) + \frac{4M\eta}{\tau}. \tag{50}$$

Summing (50) over $t = 0, \ldots, T-1$ and dividing by $T$ gives

$$\frac{1}{NT} \sum_{t=0}^{T-1} \sum_{i \in S_t} E\|\nabla f_i(\Theta_i^t)\|^2 \leq \frac{4}{\eta\tau T} \sum_{t=0}^{T-1} \frac{1}{N} \sum_{i \in S_t} \left( E[f_i(\Theta_i^t)] - E[f_i(\Theta_i^{t+1})] \right) + \frac{4M\eta}{\tau}. \tag{51}$$

Using $f_i(\Theta_i^0) - f_i^\star \leq D$ for all $i$ and averaging gives

$$\frac{1}{T} \sum_{t=0}^{T-1} \frac{1}{N} \sum_{i \in S_t} \left( E[f_i(\Theta_i^t)] - E[f_i(\Theta_i^{t+1})] \right) \leq D.$$

Plugging into (51) yields

$$\frac{1}{NT} \sum_{t=0}^{T-1} \sum_{i \in S_t} E\|\nabla f_i(\Theta_i^t)\|^2 \leq \frac{4D}{\eta\tau T} + \frac{4M\eta}{\tau}.$$

Choosing $\eta = \sqrt{\frac{D}{MT}}$ gives

$$\frac{1}{NT} \sum_{t=0}^{T-1} \sum_{i \in S_t} E\|\nabla f_i(\Theta_i^t)\|^2 \leq \frac{8}{\tau} \sqrt{\frac{DM}{T}} = O\left(\frac{1}{\sqrt{T}}\right).$$

which is (45). $\qquad\square$

# D. Computational and Memory Complexity

This appendix provides a per-round, per-LoRA-matrix accounting of server-side computation and memory. We first derive concrete FLOP and storage expressions using standard dense linear algebra primitives, instantiate them with practical values, and then extract the corresponding Big-$\mathcal{O}$ scalings. Throughout, we assume the server stores all client updates in memory.

**Notation.** Let $N$ be the number of clients, $d$ the hidden dimension, and $r \ll d$ the LoRA rank. Each client sends factors $(B_i, A_i)$ with $B_i \in \mathbb{R}^{d \times r}$ and $A_i \in \mathbb{R}^{r \times d}$. All costs below are per LoRA-injected matrix (per adapted weight matrix). If a model contains $L$ such matrices, multiply all per-matrix costs by $L$.

**FLOP model.** We use the standard dense GEMM cost:

$$\texttt{GEMM}(m \times n, \ n \times p) \ \approx \ 2mnp \ \text{FLOPs}.$$

For memory, we count floats (multiply by 4 bytes for FP32, 2 bytes for FP16/BF16).

## D.1. Case 1: Full-rank averaging + rank-$r$ refactorization (SVD)

**Step A: reconstruct and average full-rank updates.** The server forms

$$\bar{W} \ = \ \frac{1}{N} \sum_{i=1}^{N} B_i A_i \in \mathbb{R}^{d \times d}.$$

The dominant computation is reconstructing each $B_i A_i$:

$$T_{\text{recon}} \approx \sum_{i=1}^{N} 2 \, (d \times r \times d) \ = \ 2Nd^2 r. \tag{52}$$

Accumulating $N$ dense $d \times d$ matrices and scaling by $1/N$ adds lower-order $O(Nd^2)$ FLOPs compared to (52) when $r \geq 1$.

**Step B: refactorize $\bar{W}$ to rank $r$.** The dominant term for a truncated or randomized SVD to extract a rank-$r$ factorization is

$$T_{\text{svd(trunc)}} = O(d^2 r). \tag{53}$$

**Total time (per matrix, per round).** Combining reconstruction with SVD gives:

$$T_{\text{Full+SVD}} = 2Nd^2 r \ + \ O(Nd^2) \ + \ O(d^2 r), \qquad \text{truncated SVD.} \tag{54}$$

**Peak memory (per matrix).** Storing all client factors costs $2Ndr$ floats. Materializing the dense $\bar{W}$ costs $d^2$ floats. Thus:

$$M_{\text{Full+SVD}} = 2Ndr \ + \ d^2 \ + \ O(dr) \ = \ O(2Ndr + d^2). \tag{55}$$

## D.2. Case 2: Factor-wise averaging (FedAvg on factors)

**Step: average factors directly.** The server computes $\bar{B} = \frac{1}{N} \sum_i B_i \in \mathbb{R}^{d \times r}$ and $\bar{A} = \frac{1}{N} \sum_i A_i \in \mathbb{R}^{r \times d}$. Summation and scaling dominate:

$$T_{\text{Fact}} \approx (N-1) \cdot (dr) \ + \ (N-1) \cdot (rd) \ + \ O(dr) \ = \ O(Ndr). \tag{56}$$

**Peak memory (per matrix).** Under the stated assumption that all client updates are stored:

$$M_{\text{Fact}} = 2Ndr \ + \ O(dr) \ = \ O(Ndr). \tag{57}$$

## D.3. Case 3: PERSIA (global FL aggregation)

PERSIA adds three operations: (i) Gram-based whitening per client, (ii) truncated SVD on stacked whitened factors, and (iii) re-basing plus averaging.

**Step A: client-wise whitening.** For each client, form the Gram matrix $G_i = B_i^\top B_i \in \mathbb{R}^{r \times r}$ and then compute a whitening transform applied to $A_i$. The dominant dense multiplications are:

$$T_{\text{Gram}} \approx \sum_{i=1}^{N} 2\,(r \times d \times r) \;=\; 2Ndr^2, \tag{58}$$

$$T_{\text{whiten-mul}} \approx \sum_{i=1}^{N} 2\,(r \times r \times d) \;=\; 2Ndr^2. \tag{59}$$

Computing the matrix square root (or Cholesky/eigendecomposition) of an $r \times r$ SPD matrix contributes $O(Nr^3)$, which is typically smaller than $O(Ndr^2)$ when $d$ is large and $r$ is modest.

**Step B: truncated SVD on stacked whitened factors.** PERSIA stacks whitened factors into $\tilde{Z} \in \mathbb{R}^{(Nr) \times d}$ and extracts the top-$r$ right singular vectors. A standard truncated SVD (or randomized SVD) has dominant cost:

$$T_{\text{tSVD on } (Nr) \times d} = O\big((Nr)\,d\,r\big) \;=\; O(Ndr^2). \tag{60}$$

**Step C: re-basing and aggregation.** Re-basing per client requires multiplying $A_i$ by the shared basis and then updating $B_i$:

$$T_{\text{rebase}} \approx \sum_{i=1}^{N} \Big( 2\,(r \times d \times r) + 2\,(d \times r \times r) \Big) \;=\; 4Ndr^2, \tag{61}$$

followed by averaging rebased $B_i$ at cost $O(Ndr)$.

**Total time (per matrix, per round).** Collecting (58)–(61) yields:

$$T_{PERSIA} = \underbrace{(2Ndr^2 + 2Ndr^2)}_{\text{Gram + whiten-mul}} + \underbrace{O(Nr^3)}_{\text{SPD factorization}} + \underbrace{O(Ndr^2)}_{\text{tSVD on } (Nr) \times d} + \underbrace{4Ndr^2}_{\text{re-basing}} + O(Ndr)$$

$$= O(Ndr^2). \tag{62}$$

**Peak memory (per matrix).** With all client factors stored, PERSIA additionally stores the stacked whitened matrix $\tilde{Z} \in \mathbb{R}^{(Nr) \times d}$:

$$M_{PERSIA} = \underbrace{2Ndr}_{\text{store all } (B_i, A_i)} + \underbrace{Nrd}_{\text{stacked } \tilde{Z}} + O(dr + r^2) \;=\; O(Nrd). \tag{63}$$

### D.4. Big-$\mathcal{O}$ summary (derived from the accounting)

From (54)–(63) we obtain the asymptotic per-round server complexity per LoRA-injected matrix:

*Table 5.* Server-side per-round complexity per LoRA-injected matrix (storing all clients in memory).

| Method | Time Complexity | Memory Complexity |
| --- | --- | --- |
| Factor-wise averaging | $\mathcal{O}(Ndr)$ | $\mathcal{O}(Ndr)$ |
| PERSIA | $\mathcal{O}(Ndr^2)$ | $\mathcal{O}(Nrd)$ |
| Full-rank + SVD | $\mathcal{O}(Nd^2r + d^2r)$ | $\mathcal{O}(2Ndr + d^2)$ |

**Scaling across all adapted matrices.** If the model contains $L$ LoRA-injected matrices, multiply the time and memory complexities above by $L$. In particular, the per-round server time becomes $L \cdot T_{\text{method}}$, and the peak memory becomes $L \cdot M_{\text{method}}$ up to shared workspace.

# E. Additional Results on Number of local epochs

We study how the aggregation rule behaves as we increase the amount of local computation between communication rounds. We vary the number of local steps per round from 1 to 640, while keeping the total number of processed mini-batches fixed so that all settings use the same overall training budget. This sweep interpolates between frequent communication (small local steps) and near one-shot aggregation (large local steps). We summarize the results of personalization and global training in Table 6 and Table 7, respectively.

**Global FL.** Across all GLUE tasks, factor-wise averaging becomes increasingly unstable as the number of local steps grows: performance degrades sharply and often collapses for large local steps due to the low-rank aggregation mismatch in Equation (1). FedProx mitigates drift but does not eliminate the mismatch, so it also degrades as the number of local steps increases. In contrast, PERSIA remains stable across the full range of local steps and closely tracks the full-rank reference, indicating that resolving the aggregation mismatch is key to enabling long local training intervals.

**Personalized FL.** The effect is stronger in PFL because the server shares only the right factor and cannot directly access $B_i$, making the mismatch more pronounced. Baselines degrade rapidly with larger local steps, while PERSIA stays robust, preserving high accuracy even when local training is long. This supports the roles of Gram whitening and shared right-subspace recovery in making aggregation well-posed under partial sharing.

**Per-task curves.** Figs. 6 and 7 report the complete task-wise accuracy curves as a function of local steps for global FL and personalized FL, respectively. These results complement Fig. 5 in the main text and confirm that PERSIA consistently extends the feasible local-step regime without sacrificing convergence.

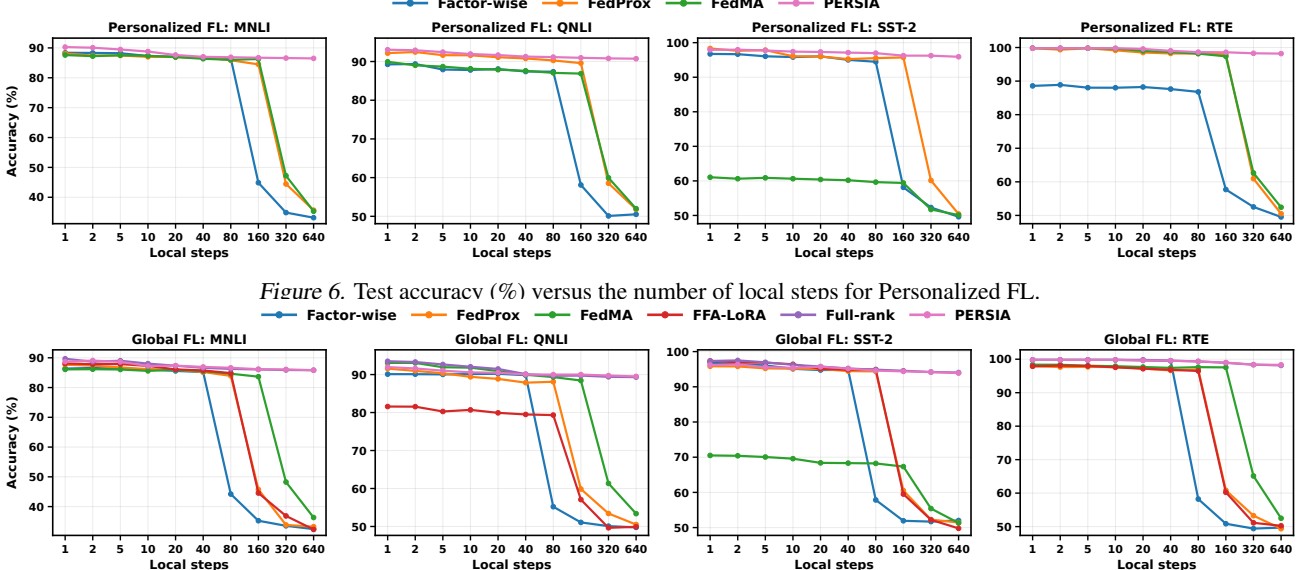

Figure 6. Test accuracy (%) versus the number of local steps for Personalized FL.

Figure 7. Test accuracy (%) versus the number of local steps for Global FL.

*Table 6.* Refined test accuracy (%) on NLU (GLUE) benchmarks under **Personalized FL**.

| Task | Method | 1 | 2 | 5 | 10 | 20 | 40 | 80 | 160 | 320 | 640 |
|------|--------|---|---|---|----|----|----|----|-----|-----|-----|
| MNLI | FedAvg | **91.55** | **90.03** | 88.51 | 87.72 | 87.52 | 86.37 | 38.58 | 34.90 | 33.72 | 33.64 |
| | FedProx | 90.80 | 89.74 | **88.95** | 87.97 | **87.81** | 86.82 | 86.09 | 39.02 | 36.61 | 32.62 |
| | FedMA | 89.97 | 89.71 | 88.57 | 88.06 | 87.01 | 86.61 | 85.97 | 84.72 | 46.14 | 42.09 |
| | PERSIA | 89.87 | 89.26 | 88.72 | **88.27** | 87.52 | **87.01** | **86.55** | **86.02** | **85.99** | **85.94** |
| SST-2 | FedAvg | 65.89 | 65.01 | 63.08 | 61.60 | 61.36 | 60.19 | 50.39 | 50.34 | 50.31 | 50.25 |
| | FedProx | 99.65 | 97.95 | 96.98 | 96.92 | 95.77 | 95.25 | 94.50 | 62.17 | 55.94 | 53.24 |
| | FedMA | 98.96 | 97.55 | 97.19 | 96.84 | 95.84 | 95.02 | 94.97 | 94.92 | 56.82 | 52.73 |
| | PERSIA | **99.95** | **99.25** | **98.79** | **97.74** | **97.71** | **97.12** | **96.90** | **96.85** | **95.48** | **95.22** |
| QNLI | FedAvg | 91.88 | 91.53 | 89.20 | 88.45 | 87.74 | 87.42 | 51.89 | 49.01 | 48.98 | 48.95 |
| | FedProx | **95.77** | **94.63** | **93.51** | **92.49** | 91.54 | 90.80 | 88.32 | 57.16 | 50.88 | 50.82 |
| | FedMA | 91.27 | 90.17 | 89.22 | 88.86 | 88.52 | 87.60 | 86.89 | 85.65 | 60.57 | 56.08 |
| | PERSIA | 93.86 | 93.60 | 92.70 | 92.31 | **91.78** | **91.27** | **90.91** | **90.86** | **90.16** | **90.05** |
| QQP | FedAvg | 74.15 | 73.20 | 71.35 | 70.44 | 70.41 | 69.15 | 52.71 | 49.00 | 48.97 | 48.94 |
| | FedProx | 78.03 | 76.62 | 75.43 | 74.74 | 73.25 | 73.06 | 71.45 | 54.32 | 54.29 | 53.86 |
| | FedMA | 77.18 | 75.68 | 74.82 | 73.54 | 72.88 | 72.11 | 69.27 | 69.22 | 59.73 | 56.13 |
| | PERSIA | **88.50** | **88.21** | **87.42** | **86.91** | **86.40** | **85.90** | **85.84** | **85.37** | **85.31** | **84.02** |
| RTE | FedAvg | 93.08 | 91.78 | 90.12 | 88.99 | 88.44 | 87.63 | 54.13 | 50.66 | 50.62 | 50.56 |
| | FedProx | 99.95 | 99.22 | 98.80 | 98.76 | 98.50 | 98.23 | 98.20 | 56.82 | 54.02 | 52.79 |
| | FedMA | 99.94 | 99.61 | 99.19 | 98.98 | 98.95 | 98.64 | 96.89 | 96.84 | 62.74 | 58.47 |
| | PERSIA | **99.95** | **99.63** | **99.43** | **99.40** | **99.35** | **99.01** | **98.95** | **97.52** | **97.48** | **97.42** |
| AVG | FedAvg | 83.31 | 82.31 | 80.45 | 79.44 | 79.10 | 78.15 | 49.54 | 46.78 | 46.52 | 46.47 |
| | FedProx | 92.84 | 91.63 | 90.73 | 90.18 | 89.38 | 88.83 | 87.71 | 53.90 | 50.35 | 48.67 |
| | FedMA | 91.46 | 90.54 | 89.80 | 89.26 | 88.64 | 88.00 | 86.80 | 86.27 | 57.20 | 53.10 |
| | PERSIA | **94.43** | **93.99** | **93.41** | **92.93** | **92.55** | **92.06** | **91.83** | **91.33** | **90.88** | **90.53** |

*Table 7.* Refined test accuracy (%) on NLU (GLUE) benchmarks under **Global FL**.

| Task | Method | 1 | 2 | 5 | 10 | 20 | 40 | 80 | 160 | 320 | 640 |
|------|--------|-----|-----|-----|-----|-----|-----|-----|-----|-----|-----|
| MNLI | FedAvg | **90.66** | **89.47** | **88.88** | 86.60 | 85.67 | 85.20 | 38.16 | 34.56 | 34.51 | 34.43 |
| | FedProx | 89.61 | 88.50 | 87.38 | 86.44 | 86.03 | 85.22 | 85.17 | 41.07 | 34.28 | 34.24 |
| | Freeze-A | 90.07 | 88.72 | 87.93 | 87.29 | 86.49 | 85.74 | 36.33 | 32.34 | 32.31 | 32.28 |
| | FedMA | 89.91 | 89.47 | 87.71 | 87.34 | 86.63 | 85.53 | 85.18 | 83.35 | 48.46 | 39.22 |
| | Full-Rank | 89.70 | 89.21 | 88.00 | **87.96** | 87.16 | 86.62 | 85.94 | 85.87 | 85.82 | 84.94 |
| | PERSIA | 89.37 | 88.90 | 88.10 | 87.52 | **87.47** | **86.93** | **86.88** | **86.22** | **86.19** | **86.14** |
| SST-2 | FedAvg | 73.18 | 72.22 | 71.03 | 69.60 | 68.99 | 68.32 | 51.17 | 51.12 | 51.08 | 51.03 |
| | FedProx | 98.51 | 97.25 | 96.75 | 95.67 | 95.43 | 94.45 | 93.19 | 59.90 | 54.66 | 54.59 |
| | Freeze-A | **99.19** | **98.67** | **97.34** | **96.30** | 95.28 | 94.83 | 57.10 | 57.04 | 53.38 | 53.32 |
| | FedMA | 98.87 | 98.23 | 97.02 | 95.87 | 95.19 | 94.81 | 94.13 | 92.76 | 56.92 | 55.80 |
| | Full-Rank | 97.81 | 97.67 | 96.63 | 96.19 | **95.81** | **95.20** | **94.43** | 94.22 | **94.17** | 93.04 |
| | PERSIA | 97.76 | 97.15 | 96.52 | 95.80 | 95.49 | 95.18 | 94.42 | **94.36** | 94.15 | **93.49** |
| QNLI | FedAvg | 92.19 | 91.89 | 90.21 | 89.81 | 88.04 | 87.87 | 52.99 | 49.01 | 48.98 | 48.95 |
| | FedProx | **94.32** | **93.48** | 91.88 | 91.19 | **90.74** | 89.84 | 87.56 | 54.80 | 50.23 | 50.17 |
| | Freeze-A | 84.73 | 83.38 | 81.69 | 81.07 | 80.42 | 79.50 | 52.13 | 51.55 | 51.47 | 51.04 |
| | FedMA | 93.37 | 92.34 | 91.81 | 90.88 | 90.02 | 89.95 | 86.70 | 86.64 | 57.96 | 57.91 |
| | Full-Rank | 93.37 | 92.54 | **92.08** | **91.36** | 90.57 | **90.12** | 89.62 | 89.18 | **88.95** | 87.91 |
| | PERSIA | 92.59 | 92.23 | 91.46 | 91.05 | 90.69 | 90.02 | **89.64** | **89.32** | 88.57 | **88.54** |
| QQP | FedAvg | 70.30 | 68.99 | 67.31 | 66.45 | 64.67 | 64.64 | 55.87 | 51.42 | 51.36 | 51.33 |
| | FedProx | 70.42 | 69.78 | 68.14 | 68.06 | 66.71 | 66.37 | 65.34 | 54.07 | 52.71 | 52.65 |
| | Freeze-A | 77.10 | 75.38 | 74.36 | 73.34 | 72.26 | 71.59 | 54.93 | 50.40 | 50.33 | 50.28 |
| | FedMA | 70.37 | 69.00 | 67.53 | 66.92 | 66.34 | 65.55 | 64.14 | 63.51 | 52.84 | 52.81 |
| | Full-Rank | **87.15** | **86.97** | **86.19** | **85.62** | **84.85** | **84.67** | **84.63** | **84.24** | **83.61** | **83.55** |
| | PERSIA | 78.06 | 77.81 | 76.55 | 76.36 | 75.43 | 75.16 | 74.26 | 74.21 | 74.15 | 73.88 |
| RTE | FedAvg | **99.95** | 99.21 | 98.23 | 97.47 | 97.22 | 96.65 | 51.21 | 50.69 | 50.65 | 50.59 |
| | FedProx | 99.95 | 99.83 | 99.79 | 99.75 | 99.56 | 99.53 | 96.49 | 63.39 | 57.97 | 55.29 |
| | Freeze-A | **99.95** | 98.87 | 98.19 | 97.38 | 97.34 | 96.85 | 58.20 | 54.40 | 53.95 | 53.90 |
| | FedMA | 99.75 | 99.07 | 98.79 | 98.12 | 97.58 | 97.41 | 97.37 | 97.32 | 61.81 | 52.10 |
| | Full-Rank | 99.93 | 99.76 | 99.72 | 99.63 | 99.60 | **99.56** | 98.95 | **98.62** | **98.54** | 97.71 |
| | PERSIA | **99.95** | **99.95** | **99.94** | **99.91** | **99.87** | **99.56** | **99.25** | 98.42 | 97.90 | **97.83** |
| AVG | FedAvg | 85.26 | 84.36 | 83.13 | 81.99 | 80.92 | 80.54 | 49.88 | 47.36 | 47.32 | 47.27 |
| | FedProx | 90.56 | 89.77 | 88.79 | 88.22 | 87.70 | 87.08 | 85.55 | 54.65 | 49.97 | 49.39 |
| | Freeze-A | 90.21 | 89.00 | 87.90 | 87.08 | 86.36 | 85.70 | 51.74 | 49.14 | 48.29 | 48.16 |
| | FedMA | 90.45 | 89.62 | 88.57 | 87.83 | 87.15 | 86.65 | 85.51 | 84.72 | 55.60 | 51.57 |
| | Full-Rank | **93.59** | **93.23** | **92.52** | **92.15** | **91.60** | **91.23** | **90.71** | **90.43** | **90.22** | **89.43** |
| | PERSIA | 91.55 | 91.21 | 90.51 | 90.13 | 89.79 | 89.37 | 88.89 | 88.51 | 88.19 | 87.97 |

