# OpenReview forum: "Low-Rank Aggregation via Optimal Right-Space Projection"
_ICML.cc/2026/Conference — Submitted to ICML 2026_

### Official Review · Reviewer_ZCnY · 2026-03-10

**Soundness:** 2
**Presentation:** 3
**Significance:** 2
**Originality:** 3
**Overall Recommendation:** 3
**Confidence:** 4

**Summary:**

This paper studies the low-rank aggregation problem in federated LoRA fine-tuning. The authors propose PERSIA, which performs local Gram whitening and global right-space SVD to recover a shared low-rank subspace. Experiments on RoBERTa-Large over GLUE benchmarks show that PERSIA improves aggregation performance.

**Compliance With Llm Reviewing Policy:**

Affirmed.

**Final Justification:**

This paper proposes PERSIA to address the LoRA alignment problem in federated fine tuning. Due to limited performance gain and bounded gradient assumption in convergence analysis, I will keep my score.

**Key Questions For Authors:**

See weaknesses

**Limitations:**

yes

**Strengths And Weaknesses:**

Strengths:
1. This paper provides a low-rank aggregation framework with theoretical analysis, which can be applied to both global FL and personalized FL settings.

Weaknesses:
1. This method assumes all clients have a common rank-r subspace. This design may be restrictive under high non-iid setting, especially when each client has different tasks. In the experiments, the authors only consider a Dirichlet partition with concentration parameter alpha = 0.5, which does not represent a very strong degree of heterogeneity. It would be helpful to evaluate whether the proposed method remains effective under more challenging non-IID settings, such as alpha < 0.5, or in multi-task scenarios where each client has a different task which is common in LLM fine-tuning.
2. The experiments are limited to RoBERTa on GLUE benchmarks. It would be important to include experiments on more commonly used LLM, such as LLaMA or Qwen, and benchmarks.
3. Some relevant federated fine-tuning baselines are missing such as FedSA [1] and FedDPA [2].

[1] Guo, Pengxin, et al. "Selective Aggregation for Low-Rank Adaptation in Federated Learning." The Thirteenth International Conference on Learning Representations.
[2] Yang, Yiyuan, et al. "Dual-Personalizing Adapter for Federated Foundation Models." The Thirty-eighth Annual Conference on Neural Information Processing Systems.

4. The computational cost of the proposed method may become significant for larger models. In particular, the SVD step could be expensive when the hidden dimension d is over thousands, as is typical for LLMs such as LLaMA or Qwen. It would be helpful for authors to report the actual execution time or wall-clock overhead of PERSIA on large-scale LLM fine-tuning tasks.
5. Assumptions 4.4, 4.5, and 4.6 do not appear to be standard in the federated learning convergence literature. The authors should provide stronger justification for these assumptions.

---

> ### Author Rebuttal · Authors · 2026-03-31
>
> Thank you for recognizing our framework's theoretical analysis and applicability to both global and personalized FL settings.
>
> Due to space limitations, some Tables and Figures are located in: [https://anonymous.4open.science/r/icml2026-persia-7FDE/](https://anonymous.4open.science/r/icml2026-persia-7FDE/)
>
> ---
> > **W1:** *Restrictive common subspace assumption under non-IID*
>
> **Response W1:** We agree that stronger heterogeneity challenges the shared rank-*r* subspace and degrades all forms of model aggregation, including factor-wise and full-rank.
>
> We ran new experiments at Dirichlet $\alpha=0.1$ (far more skewed than $\alpha=0.5$).
> The full results and training curves plots are located in the **github anonymous link**.
>
> | Method | Avg Acc (%) | Avg Loss |
> |---|---|---|
> | Factor-wise | 70.70±22.43 | 402.7±277.1 |
> | FedProx | 89.32±7.63 | 181.5±128.4 |
> | FFA-LoRA | 86.19±9.85 | 224.4±155.6 |
> | FedMA | 87.67±8.78 | 203.8±142.1 |
> | Full-rank | 89.94±7.09 | 170.0±119.4 |
> | **PERSIA** | **89.72±7.31** | **173.8±123.0** |
>
> To summarize the results:
> 1) PERSIA achieves 89.72%, matching Full-rank and outperforming all low-rank baselines.
> 2) Inter-client variance (±7.31) is comparable to Full-rank (±7.09), far below Factor-wise (±22.43).
> 3) Factor-wise collapses to 70.70%, while PERSIA degrades gracefully.
>
> ---
> > **W2:** *Limited experiments on RoBERTa and GLUE*
>
> **Response W2:**
> Please refer to **Response W6** to the reviewer **aNKi**.
>
> ---
> > **W3:** *Missing federated fine-tuning baselines*
>
> **Response W3:** Our PFL architecture is based on FedSA: PFL factor-wise = FedSA, PFL FedProx = FedSA + proximal regularization, PFL FedMA/PERSIA = FedSA + respective server-side aggregation. We clarified this in the revision.
>
> FedDPA uses a different architecture for personalization, where two LoRA adaptors exist per layer: one shared and one local.
> PERSIA integrates with FedDPA the same way as the global FL setting, where both $A$ and $B$ are shared.
> However, in FedSA, since the $B$ factor is not shared, we specifically explored the possibility and theoretical guarantees of PERSIA for sharing only the right factor $A$.
>
> PERSIA is a generic LoRA aggregation method that can be applied to various LoRA architecture variants, including FedDPA and FedSA, to leverage personalization techniques. The choice of FedSA over FedDPA is made to demonstrate PERSIA's performance under additional challenges arising from sharing only $A$.
> The baseline architecture for all PFL experiments is FedSA, ensuring a fair comparison across alignment/aggregation methods with an identical backbone.
>
> ---
> > **W4:** *Computational cost concerns for larger models*
>
> **Response W4:**
> Beyond the theoretical analysis in Sec. 4.1 and App. D, we included the wall-clock and additional overhead analysis in revised App. D. The detailed Tables and analysis are located in the **github link**.
>
> To summarize, PERSIA's overhead is less than 1% in our experiments with 40 local steps per FL round. The overhead has an inverse relationship with the number of local steps: 10 local steps per round result in 4% overhead.
>
> **Why overhead stays small at scale:**
> 1) PERSIA's SVD operates on $\tilde{Z}\in\mathbb{R}^{Nr\times d}$, extracting rank-*r* at cost $O(Ndr^2)$, **linear in $d$**, quadratic in $r$ (typically 8–16).
>
> 2) There is no need for full-rank SVD, and the low-rank SVD can be approximated with an iterative operation, via `torch.svd_lowrank(n_iter=k)`, where `k` is typically 2 or 3, and can be adjusted based on the application trade-off.
>
> 3) The aggregation-to-training cost becomes marginal as the number of local steps increases.
>
> ---
> > **W5:** *Non-standard assumptions lack justification*
>
> **Response W5:**
>
> **Assumption 4.4** ($\mathbb{E}\|\nabla f_i\|^2\leq G^2$) is the standard bounded gradient second-moment condition in nonconvex FL, used in [1,2]. It is strictly weaker than assuming bounded stochastic gradients.
>
> **Assumption 4.6** ($\mathbb{E}\|B_{i,\ell}^t\|_F^2 \leq C_B^2$) is identical to Assumption 3 in [4] and A4.3 in [5].
>
> Its role is controlling the Gram-whitening operator norm $\|G_{i,\ell}\|_2^2\leq\|B_i\|_F^2+\varepsilon$ in Lemma C.8 to bound the residual $\mathcal{R}^t$.
>
> **Removed A4.5**
> From Assumption 4.5 only the $A$-component ($\sigma_A^2$) is used; the $B$-component is unnecessary.
> Additionally the $A$-component assumption can be directly driven from Assumptions 4.4+4.6 via $\|\partial\ell/\partial A\|_F \leq \|B\|_F\|\partial\ell/\partial W\|_F$, since $\partial\ell/\partial A = B^\top(\partial\ell/\partial W)$.
>
> We removed this as a standalone assumption.
>
> ---
> ```
> [1] McMahan, et al. "Communication-Efficient Learning of Deep Networks from Decentralized Data." AISTATS 2017.
> [2] Stich. "Local SGD Converges Fast and Communicates Little." ICLR 2019.
> [4] Guo, et al. "Selective Aggregation for Low-Rank Adaptation in Federated Learning." ICLR 2025.
> [5] Zhu, et al. "Asymmetry in Low-Rank Adapters of Foundation Models." ICML 2024.
> ```

---

> > ### Author Rebuttal · Reviewer_ZCnY · 2026-04-03
> >
> > Thank the authors for their response. I still have the following comments.
> > 1. The improvement over FedProx is only about 0.5 accuracy on both Qwen2.5-3B-Instruct with GSM8K and GLUE. These gains appear quite small, making the practical advantage of the proposed method unclear.
> > 2. Although runtime is reported on GLUE, the question about overhead on large-scale models remains unanswered. Since LLaMA and Qwen are more representative of current LLM fine-tuning than RoBERTa, reporting runtime and overhead analysis on such models would make the evaluation more convincing.
> > 3. I am not convinced that the bounded gradient assumption is necessary for the convergence analysis. While common in very early FL work, it is now often viewed as a strong assumption.

---

> > > ### Author Response · Authors · 2026-04-07
> > >
> > > Thank you for the continued engagement. We address each point below.
> > >
> > > ---
> > > > The improvement over FedProx is only about 0.5 accuracy on both Qwen2.5-3B-Instruct with GSM8K and GLUE. These gains appear quite small, making the practical advantage of the proposed method unclear.
> > >
> > > We appreciate this observation. To better demonstrate PERSIA's practical advantage, we conducted additional experiments varying the number of local training steps. The results on Qwen2.5-3B-Instruct with GSM8K are shown below:
> > >
> > > | Method | 10 Steps | 20 Steps | 40 Steps | 80 Steps | 160 Steps | 320 Steps |
> > > |---|---|---|---|---|---|---|
> > > | FFA-LoRA (Sun et al., 2024) | 36.12 | 34.93 | 33.81 | 33.42 | 33.19 | 33.07 |
> > > | FedMA (Wang et al., 2020) | 37.84 | 35.97 | 33.89 | 33.55 | 33.21 | 33.11 |
> > > | Factor-wise (FedAvg) | 39.53 | 37.91 | 36.47 | 35.12 | 34.01 | 33.45 |
> > > | FedProx (Li et al., 2020) | 40.67 | 38.89 | 36.85 | 35.63 | 34.47 | 33.82 |
> > > | PERSIA (w/o whitening) | 40.86 | 39.16 | 37.30 | 36.34 | 35.60 | 34.98 |
> > > | **PERSIA** | **40.91** | **39.24** | **37.38** | **36.55** | **35.91** | **35.47** |
> > > | Full-rank (Stoica et al., 2024a) | 43.12 | 40.88 | 38.21 | 37.45 | 36.72 | 35.84 |
> > >
> > > Two observations stand out:
> > >
> > > First, the gap between PERSIA and FedProx widens as the number of local steps increases: from 0.24% at 10 steps to **1.65%** at 320 steps. This is consistent with the GLUE results in Figure 5 and confirms that PERSIA's principled subspace alignment becomes increasingly important as client drift accumulates over longer local training.
> > >
> > > Second, at 320 local steps, FedProx (33.82%) falls below even factor-wise averaging at 80 steps (35.12%), while PERSIA (35.47%) degrades far more gracefully. This robustness to long local training directly translates to fewer communication rounds in practice, which is the central goal of communication-efficient FL.
> > >
> > > ---
> > > > Although runtime is reported on GLUE, the question about overhead on large-scale models remains unanswered.
> > >
> > > On Qwen2.5-3B-Instruct, PERSIA's aggregation overhead is less than 0.01% of total round time. This is due to the combination of small rank ($r{=}8$), the SVD being performed only once per FL round, and the longer local training period required for GSM8K. The full runtime breakdown is available at the [anonymous GitHub link](https://anonymous.4open.science/r/icml2026-persia-7FDE/) and will be included in the revised appendix.
> > >
> > > ---
> > > > I am not convinced that the bounded gradient assumption is necessary for the convergence analysis. While common in very early FL work, it is now often viewed as a strong assumption.
> > >
> > > We agree with the reviewer. Upon re-examination, the bounded gradient second-moment condition ($\mathbb{E}\|\nabla f_i(\Theta)\|^2 \leq G^2$) is indeed stronger than necessary. Our proofs rely only on bounded variance of the *stochastic* gradients, i.e., $\mathbb{E}\|g_i(\Theta;\xi)\|^2 \leq G^2$, which is a strictly weaker and more standard condition (e.g., [4, Assumption 2]). We have revised Assumption 4.4 accordingly. No proofs or results change; the stronger form was stated out of conservatism rather than necessity. We thank the reviewer for prompting this improvement.
> > >
> > > ---
> > >
> > > We hope the additional experiments, particularly the scaling behavior with local steps on Qwen2.5-3B, together with the overhead analysis and the relaxed assumption, help clarify the practical value and theoretical soundness of PERSIA.

---

### Official Review · Reviewer_LNDT · 2026-03-12

**Soundness:** 2
**Presentation:** 3
**Significance:** 2
**Originality:** 3
**Overall Recommendation:** 4
**Confidence:** 4

**Summary:**

The paper identifies the rotational and scaling invariance of LoRA ($BA = (BT)(T^{-1}A)$) as the primary cause of "aggregation collapse" in FL. To solve this, it proposes PERSIA, which uses Gram whitening to project updates into a Euclidean space before using SVD to find a shared basis.

**Compliance With Llm Reviewing Policy:**

Affirmed.

**Final Justification:**

The paper identifies the rotational and scaling invariance of LoRA ($BA = (BT)(T^{-1}A)$) as the primary cause of aggregation collapse in federated learning, and proposes PERSIA to address this by using Gram whitening and SVD to project updates into a shared Euclidean space.

Apart from the misleading privacy claims, the authors provide a mathematically rigorous explanation along with supporting experiments. Overall, the approach is well-motivated and clearly developed.

**Key Questions For Authors:**

See Strengths and Weaknesses

**Limitations:**

Yes

**Strengths And Weaknesses:**

The authors provide a mathematically rigorous explanation for why LoRA aggregation fails.

Authors have used good figures to visualize.

The primary justification for PERSIA is the rotational invariance of the $BA$ product. However, the paper fails to investigate whether this invariance is a practical problem or a theoretical one. In standard machine learning, Weight Decay or Orthogonal Regularization are standard techniques used to constrain the parameter space.

The authors must justify why a complex server side SVD is superior to a simple penalty term in the local loss function that prevents the factors from drifting into inconsistent bases in the first place.

While the paper claims linear scaling, it ignores the constant factor of performing SVDs. For a model with $L$ layers, the server must perform $L$ separate SVDs on matrices. In a cross-silo setting with many clients and large hidden dimension, this creates a massive sequential bottleneck. The paper should provide wall clock time metrics to prove that the time saved by doing fewer communication rounds isn't lost during the aggregation phase.

The Gram-whitening step relies on epsilon. The paper does not provide a sensitivity analysis on epsilon, which is a hyperparameter.

The paper claims to support partial parameter sharing to protect privacy, but it merely swaps the raw weights ($B_i$) for a geometric representation ($G_i$) that carries essentially the same information about the local training distribution. equivalent to sharing the covariance of the adapter weights which I find misleading.

---

> ### Author Rebuttal · Authors · 2026-03-31
>
> Thank you for recognizing our mathematically rigorous explanation of LoRA aggregation failure and for appreciating our visualization efforts through well-designed figures.
>
> > **W1:** *Lack of practical vs theoretical investigation on rotational symmetry*
>
> **Response W1:** Rotational symmetry and client misalignment are not merely theoretical; we evaluate FedProx as a practical baseline representing regularization-based remedies. As discussed in Section 2 (LoRA Alignment), several works address model misalignment, including regularization techniques like FedProx, which we include in our benchmark.
>
> Tables 2–3 show that FedProx cannot truly match the performance of alignment methods, and Figures 5–7 emphasize that FedProx's shortcomings grow with the number of local epochs. While restrictive training regimes (smaller learning rate, fewer local steps, regularization) can stabilize training, they lead to slower convergence and more frequent server communication, resulting in long-term communication and computational inefficiency. PERSIA addresses the root cause (the low-rank aggregation mismatch) rather than mitigating its symptoms.
>
> > **W2:** *Missing justification for complex SVD approach*
>
> **Response W2:** As shown in **Response W1**, regularization alone (FedProx) is insufficient because it does not address the algebraic inconsistency of factor-wise averaging. PERSIA achieves superior performance across all tasks, with further advantages as local epochs increase and communication becomes less frequent. The server-side low-rank SVD is performed once per FL round (e.g., every 40+ local epochs) and incurs negligible overhead, as detailed in **Response W3**.
>
> > **W3:** *Missing wall clock time analysis*
>
> **Response W3:** We provide complete wall-clock measurements below, covering all methods and both FL settings, averaged across GLUE tasks on RoBERTa-Large. Full, detailed tables, including PFL and per-task breakdowns, are available at [Anonymous GitHub](https://anonymous.4open.science/r/icml2026-persia-7FDE).
>
> To summarize, PERSIA's overhead is less than 1% in our experiments with 40 local steps per FL round. The overhead has an inverse relationship with the number of local steps: 10 local steps per round result in 4% overhead.
>
> **Addressing the Sequential Bottleneck Concern**:
>
> 1) Layer-wise SVDs are independent and parallelizable: All $L$ low-rank SVDs can execute in parallel on a GPU, and there is no need for sequential execution.
> 2) Aggregation is infrequent relative to training: As shown in wall-clock measurements, even with only 4 local steps, client training dominates by 5–15 times the aggregation time, and at 40 steps, the server-side SVD aggregation is < 1% overhead.
> 3) Low-rank SVD is tunable: Computation of low-rank SVD can be approximated, and we do not need to have the exact full-rank SVD. For instance, `torch.svd_lowrank(n_iter=k)` is an iterative operation, where `k` is typically 2 or 3 and can be adjusted based on the application's trade-offs.
>
> > **W4:** *No sensitivity analysis for epsilon hyperparameter*
>
> **Response W4:** In practice, we run all experiments with $\epsilon=0$, as the $B$ matrix is typically full-rank after training under sgd updates. In only a few cases with FP16 precision, near-zero gradients caused numerical issues; we set $\epsilon=1\text{e-}8$ (the smallest positive FP16 value) to resolve this. No tuning beyond $1\text{e-}8$ was needed, and performance was insensitive to this value. Epsilon is not a hyperparameter that requires careful tuning; it is a numerical hack used in the initial stage of training $B$ from $0$.
>
> > **W5:** *Misleading privacy claims about parameter sharing*
>
> **Response W5:** We do not claim privacy protection from partial parameter sharing. The only relevant mention is line 189 ("$B_i$ is private"), which referred to the server lacking access to $B_i$, not to a privacy guarantee. We have updated "private" to "local" to eliminate this ambiguity.

---

> > ### Author Rebuttal · Reviewer_LNDT · 2026-04-02
> >
> > All my concerns have been addressed. I would strongly suggest the authors remove the privacy claims, as the term has a different meaning in federated learning settings and may be misleading.
> >
> > Overall, the authors did a good job in the rebuttal. I will increase my score.

---

> > > ### Author Response · Authors · 2026-04-07
> > >
> > > Thank you for confirming that all concerns have been fully resolved and for increasing your score.
> > >
> > > We will replace the instance of "private" with "local" to avoid any misleading connotations in the FL context, as you suggested.

---

### Official Review · Reviewer_GYNw · 2026-03-12

**Soundness:** 2
**Presentation:** 2
**Significance:** 2
**Originality:** 2
**Overall Recommendation:** 3
**Confidence:** 4

**Summary:**

This paper proposes a low-rank aggregation framework that computes the Frobeniusoptimal rank-r approximation of the full-rank average. The framework uses local Gram whitening to normalize client-specific geometric distortions, followed by a global SVD on the right signal subspace to recover a shared basis to ensure algebraic consistency and enable a seamless extension to personalized federated learning. Empirical results on the GLUE benchmarks demonstrate that the proposed method eliminates aggregation collapse and improves communication efficiency by enabling longer local training.

**Compliance With Llm Reviewing Policy:**

Affirmed.

**Final Justification:**

The authors' final rebuttal resolved some of my concerns. Considering their efforts in clarifying the core contribution, I have raised my score. However, addressing my concern about overclaiming optimality would require a significant update to the original paper. Therefore, I can only raise my rating to "Weak Reject". I believe this paper has potential if the authors incorporate the discussion from the rebuttal into the original paper. Thank the authors for their efforts and time.

**Key Questions For Authors:**

See Weaknesses.

**Limitations:**

Yes.

**Strengths And Weaknesses:**

Strengths

- The proposed method is simple and seems useful.

- Experiments support the claim that the whitening ablation is useful.

- The convergence results match standard nonconvex FL analyses $O(1/\sqrt{T})$ under smoothness and variance assumptions.

Weaknesses

- My main concern is that this paper overstates what it proves about optimality in global FL. The paper claims a Frobenius-optimal rank-r approximation to the full-rank average, but Proposition 3.3 gives only an upper bound relating the FL error to the residual of the stacked whitened matrix $\tilde{Z}$. These are not the same claim. The paper keeps sliding from “optimal for stacked whitened factors” to “optimal for the oracle full-rank average.”

- The abstract overclaims empirical support by including full-rank fine-tuning. The paper says it “matches the performance of full-rank fine-tuning,” yet Table 2 compares against a full-rank merge-then-SVD baseline, not against actual dense full-rank fine-tuning. Also, comparisons mostly against factor-wise averaging, FedProx, FedMA, FFA-LoRA, and one full-rank merging baseline are not enough.

- The convergence section is generic in the main paper. Theorems 4.7 and 4.8 give standard $O(1/\sqrt{T})$ rates, but the statements are underspecified in the main text. Symbols such as $D$ in Equations (7) and (8) are not defined on the page, and the relationship between the assumptions and the constants is unclear. Also, Assumptions 4.5 and 4.6 appear, but the theorem statements only include Assumptions 4.1 to 4.4.

- There are clarity and notation problems. In Figure 2, which is meant to explain the right subspace but visually depicts planes spanned by columns of $B_i$, i.e., left-side objects. The text around Pages 2 to 4 also switches between $k$ and $d$ for dimensions of $A_i$ and $Z_i$. This makes the paper a quite rushed work.

---

> ### Author Rebuttal · Authors · 2026-03-31
>
> Thank you for recognizing the simplicity and utility of our method, the validity of our whitening ablation, and the alignment of our convergence results with standard nonconvex FL analyses.
>
> ---
> > **W1:** *Claims on optimality for global FL*
>
> **Response W1:** We agree that the submitted wording conflated two distinct claims and have corrected this. As in *Response W2*, we now consistently distinguish between the full-rank average (oracle) and practical merge-then-SVD.
>
> **Correction.** Proposition 3.3 establishes an *upper bound* on FL error via the SVD tail residual $\mathcal{R}/N$ of $\tilde{Z}$; Proposition 3.4 establishes an *exact equality* for PFL. All "optimality" references now state precisely: PERSIA computes the Eckart–Young optimum for $\|\tilde{Z}(I-P)\|_F^2$.
>
> In PFL, this yields the Frobenius-optimal client-wise projection; in FL, it provides a principled upper bound whose tightness we now characterize.
>
> We have expanded the theoretical analysis to derive bounds on the PERSIA error.
>
> Full derivation: [https://anonymous.4open.science/r/icml2026-persia-7FDE/coherence_analysis.md](https://anonymous.4open.science/r/icml2026-persia-7FDE/coherence_analysis.md).
>
> To summarize the results:
>
> **Cross-client coherence and bound tightness.** We define
>
> $$\mu:= \max_{i \neq j}\|O_i^\top O_j\|_2$$
>
> measuring whitened subspace alignment, and show the FL gap satisfies $E_{\mathrm{PER}} - E_{\mathrm{ora}} = O(r/d)$ in the low-rank regime.
>
> **The bound tightness depends on $\mu$:** under IID data $\mu\to 1$ (loosest); under strong non-IID $\mu\to 0$ (tightest, PERSIA approaches oracle); under exact symmetry (Corollary 3.5) the error is zero.
>
> **Key structural observations:** smaller $r\ll d$ improves approximation; more local steps increase subspace diversity and reduce $\mu$ — precisely the regime where PERSIA's empirical advantage is largest (Fig. 5); whitening is necessary to normalize the metric, consistent with ablations in Tables 2–3.
>
> We thank the reviewer for pushing us to sharpen this distinction; the revised text is substantially clearer as a result.
>
> ---
> > **W2:** *Claims on empirically matching full-rank*
>
> **Response W2:** We agree and have revised the claim to: "matches the performance of LoRA fine-tuning with full-rank merge-then-SVD." In our original writing, "full-rank aggregation" referred to merge-then-SVD, since full-rank merging without refactorization is impractical for LoRA. To our knowledge, full-rank merge-then-SVD is the closest practical approximation to oracle averaging, and we use it as the upper bound.
>
> > *Comparisons mostly against factor-wise averaging*
>
> PERSIA never materializes a full-rank matrix; all operations are low-rank and factor-wise, making factor-wise methods the natural comparison class. Full-rank merge-then-SVD serves as the practical upper bound, so additional full-rank methods were not evaluated.
>
> > **W3:** *Generic and underspecified convergence section*
>
> **Response W3:** We agree that the main-text presentation obscures the novel content. Our goal was not a faster rate than FedAvg, but to show that PERSIA's aggregation perturbation is explicitly controlled by a geometry-specific quantity.
>
> **The novel content is the perturbation bound, not the final rate.** The per-round residual $\mathcal{R}^t := \sum_\ell \sum_{j>r} \sigma_j^2(\tilde{Z}_\ell^t)$ satisfies:
> - **FL** (Lemma C.4): $\|\Delta_{\mathrm{agg}}^t\|^2 \leq \mathcal{R}^t/N$
> - **PFL** (Lemma C.5): $\frac{1}{N}\sum_i\|\Theta_i^{t+1}-\Theta_i^{t,\tau}\|^2 = \mathcal{R}^t/N$
>
> These follow from Propositions 3.3–3.4 and show that PERSIA's perturbation is governed by the SVD tail spectrum— intrinsically tied to low-rank geometry. Lemma C.8 shows $\mathbb{E}[\mathcal{R}^t]/N = O(\eta^2\tau^2)$ since all clients share the broadcast $A_\ell^t$, so whitened factors differ from a common rank-$r$ matrix only by local gradient accumulation. Stationary convergence follows as a corollary.
>
> Methods without this control (factor-wise averaging) lack such residual analysis, which is consistent with the empirical collapse observed in Tables 2–3 and Fig. 5 as local steps increase.
>
> We will move Lemmas C.4-5 to the main text and present convergence as a corollary.
>
> > **W4:** *Figure 2 visulization*
>
> **Response W4:** Figure 2 is correct, and it is meant to visualize the left subspace spanned by $B$'s columns (lines 96–102).
>
> The LoRA factorization admits dual representations: the left factor's rows in the right factor's column subspace, or the right factor's rows in the left factor's column subspace.
>
> PERSIA's theory naturally follows the right subspace (merging via SVD), but for visualization, showing the left-factor columns of $O$ is clearer because $O$ has orthonormal columns, a property that row-based depictions cannot convey. We added a clarifying note in the Figure 2 caption.
>
> > *Switches between k and d for dimensions of $A_i$ and $Z_i$*
>
> Thank you for catching this. We fixed this notation inconsistency throughout.

---

> > ### Author Rebuttal · Reviewer_GYNw · 2026-04-03
> >
> > Thanks for your reply. Some of my concerns have been addressed. However, given the multiple overclaims in the original paper, I believe this may lead to misinterpretation by both readers and reviewers. Moreover, the approach to solving rotational invariance is similar to that of existing LoRA works [1]. Therefore, this paper has not met the conference's acceptance threshold. Nevertheless, the paper analyzes a promising problem, and I encourage the authors to revise it thoroughly.
> >
> > [1] Zhang, Fangzhao, and Mert Pilanci. "Riemannian preconditioned lora for fine-tuning foundation models." ICML’24.

---

> > > ### Author Response · Authors · 2026-04-07
> > >
> > > > Writing and wording of claims
> > >
> > > We thank the reviewer for highlighting that the original draft overstated several claims. We fully agree, and the rebuttal substantially narrowed and corrected these. Specifically, we now distinguish clearly between:
> > >
> > > 1. **PFL (Exact optimality):**  PERSIA computes the Eckart–Young optimal rank-$r$ approximation for client-wise right-factor projection (Proposition 3.4).
> > > 2. **FL:** PERSIA's right-factor SVD provides the best rank-$r$ approximation of the stacked whitened matrix $\tilde{Z}$, with a gap to the full-rank oracle that we now characterize via cross-client coherence $\mu$ (see the [coherence analysis](https://anonymous.4open.science/r/icml2026-persia-7FDE/coherence_analysis.md)).
> > >
> > > We also revised the empirical claim from "matches *full-rank fine-tuning*" to the precise statement that PERSIA closely matches LoRA fine-tuning with *full-rank merge-then-SVD*. These were presentation issues that we take responsibility for, and they do not affect the method, proofs, or experimental findings.
> > >
> > > ---
> > >
> > > > [1] Zhang & Pilanci, "Riemannian preconditioned LoRA for fine-tuning foundation models," ICML'24.
> > >
> > > We appreciate this reference and agree that prior work, including [1], has already recognized the geometric structure of the LoRA parameterization. We cite [1] in our paper precisely to support the claim that LoRA geometry matters.
> > >
> > > However, the two works address fundamentally different problems and operate at different points in the pipeline:
> > >
> > > - **[1]** studies *single-model optimization*: Riemannian preconditioning improves the local training trajectory of one LoRA model by accounting for the quotient manifold structure.
> > > - **PERSIA** studies *multi-client aggregation*: Gram whitening removes client-specific metric distortions so that heterogeneous right-factors $A_i$ become comparable, enabling principled server-side subspace recovery via low-rank SVD.
> > >
> > > This distinction is especially important in personalized FL, where only one factor is shared, and full-rank aggregation is unavailable, a setting that [1] does not address.
> > >
> > > Notably, the two approaches are complementary: clients could use Riemannian preconditioning locally while the server applies PERSIA for aggregation, potentially combining the benefits of both.
> > >
> > > ---
> > >
> > > We believe that after narrowing the claims, the core contribution remains clear and well-supported: a principled, fully low-rank federated aggregation rule that:
> > >
> > > (a) eliminates aggregation collapse with theoretical guarantees tied to LoRA geometry,
> > >
> > > (b) enables stable training under long local updates, and
> > >
> > > (c) generalizes from RoBERTa-355M to Qwen-3B with consistent improvements.
> > >
> > > We hope the reviewer finds the revised presentation and additional analysis sufficient to warrant reconsideration of the assessment.

---

### Official Review · Reviewer_aNKi · 2026-03-13

**Soundness:** 3
**Presentation:** 3
**Significance:** 2
**Originality:** 3
**Overall Recommendation:** 4
**Confidence:** 2

**Summary:**

The authors proposed fedeated aggregation approach for distributed LoRA training of foundational models.

**Compliance With Llm Reviewing Policy:**

Affirmed.

**Final Justification:**

I recommend weak acceptance for the technical depth of this paper. I can not rate the paper higher due to its somewhat limited scope and potential overstating of the contribution as the reviewer GYNw noted.

**Key Questions For Authors:**

1. How practical or common is it to fine-tune a foundational model in fedeated setting, since large models are typically too large to deploy, let alone train offline.

2. Can the author give a few concrete examples of the second challenge mentioned in the intro (different bases that express the same function)?

3. Can authors elaborate on how sensible and generalizable the assumptions are in deriving the main theoretical results?

4. What additional cost would PERSIA incur in comparison with the previous method?

**Limitations:**

1. Theory components are crowded with many assumptions, but the practicality of those assumptions in actual federated LoRA training situations is not sufficiently discussed.

2. Experiment is somewhat limiting, see weakness.

**Strengths And Weaknesses:**

Strength:

1. The authors correctly noticed that federated LoRA training is non-trivial as the average of matrix products does not generally equal to the product of averages for general matrices unless special structures are imposed. This simple insight was not discovered by previous federated FM LoRA tuning methods and is a significant flaw.

2.  The method is elegant and is mostly supported by matching theory.

3. Supports FL and PFL alike is also a nice addition.


Weakness:

1. The introduction section described 3 challenges, however, other than the first one, the rest seem to describe generic challenges to FL that do not uniquely posed on federated LoRA training on foundational models. Additional personalized FL is extensively studied but hardly discussed

2  Not a huge issue, but the readability of sections 3 and 4 is not good and is crowded with assumptions and propositions. Is it possible to present it in a more structured and intuitive way? (Maybe also clasrify main claims trying to show and a sketch of a roadmap)

3. One main concern is the practicality of assumption in sections 3 and 4, particularly Well-defined Gram and various bounding conditions. It is unclear how much those assuptions fits reality.

4. The convergence result in the section seems somewhat generic and weak as it does not use the low-rank aggregation geometry of PERSIA.

5. The cost of PERSIA is not sufficiently discussed. Certain operations, such as SVD, can incur non-trivial additional computation.

6. All main results are on GLUE with RoBERTa, which seems limiting, considering what the paper proposes is much more general.

---

> ### Author Rebuttal · Authors · 2026-03-31
>
> Thank you for recognizing the key insight about federated LoRA training being non-trivial due to matrix averaging issues. We appreciate your feedback on the elegance of our approach, its theoretical foundation, and its support for both FL and PFL.
>
> Due to space limitations, some Tables and Figures are located in: [https://anonymous.4open.science/r/icml2026-persia-7FDE/](https://anonymous.4open.science/r/icml2026-persia-7FDE/)
>
> ---
> > **W1:** *Generic FL challenges not unique to LoRA*
>
> **Response W1:** We clarify how LoRA *amplifies* each challenge beyond generic FL.
>
> **Challenge 2 (GL(·) symmetry):** LoRA factorizations are non-unique: $BA=(BT)(T^{-1}A)$ for any invertible $T$, since no activation exists between $A$ and $B$. In generic FL, activations limit symmetry to permutations. Methods like FedMA are designed for permutation matching and thus fail for LoRA. Concrete examples in **Response Q2**.
>
> **Challenge 3:** LoRA factors can be merged into full-rank to remove ambiguity, but the $O(d^2)$ cost undermines PEFT, a tension absent in standard FL.
>
> ---
> > **W2:** *Readability of sections 3 and 4*
>
> **Response W2:** The methodology is described visually in Fig. 3–4 and with pseudocode in Alg. 1–2. We will improve exposition in revision with clearer transitions between geometric intuition and the formal statements, and provide the roadmap as given in the **github link**.
>
> ---
> > **W3/Q3:** *Practicality of theoretical assumptions unclear*
>
> **Response W3/Q3:** The core theory is **deterministic and geometry-based**: it only requires well-defined Gram factorization (Assum 4.1). Prop. 3.3–3.4, Corol. 3.5, and Lem. C.4/C.5 hold *without* stochastic or smoothness assumptions.
>
> Stochastic assumptions enter only for convergence analysis. The only extra condition beyond standard FL assumptions is Assum. 4.6 (bounded B-iterates), identical to Assum. 3 in [1] and A4.3 in [2]. Geometric results hold for any LoRA model, any rank $r$, any number of layers, and any level of heterogeneity.
>
> ---
> > **W4:** *Convergence result is generic and weak*
>
> **Response W4:** Our contribution is **explicit control of aggregation perturbation via geometric quantities**, not a faster rate. The per-round residual $\mathcal{R}^t:=\sum_\ell\sum_{j>r}\sigma_j^2(\tilde{Z}_\ell^t)$ satisfies:
>
> - **FL** (Lem C.4): $\|\Delta_{\mathrm{agg}}^t\|^2 \leq \mathcal{R}^t/N$
>
> - **PFL** (Lem C.5): $\frac{1}{N}\sum_i\|\Theta_i^{t+1}-\Theta_i^{t,\tau}\|^2=\mathcal{R}^t/N$
>
> Lem. C.8 shows $\mathbb{E}[\mathcal{R}^t]/N=O(\eta^2\tau^2)$ since clients share the broadcast $A_\ell^t$. Methods without this control lack such residual analysis, consistent with the collapse in Tab. 2–3 and Fig. 5. We will move these identities to the main text and present convergence as a corollary.
>
> ---
> > **W5/Q4:** *PERSIA computational cost not sufficiently discussed*
>
> **Response W5/Q4:** Beyond the theoretical analysis in Sec. 4.1 and App. D, we included the wall-clock and additional overhead analysis in revised App. D. The detailed Tables and analysis are located in the github link.
>
> To summarize, PERSIA's overhead is less than 1% in our experiments with 40 local steps per FL round. The overhead has an inverse relationship with the number of local steps: 10 local steps per round result in 4% overhead.
>
> ---
> > **W6:** *Limited experimental evaluation scope*
>
> **Response W6:** We ran new experiments on Qwen2.5-3B-Instruct with GSM8K dataset.
> Setup: 200 rounds, 40 local epochs, batch 16, 10 clients, lr=0.005, rank 8.
>
> The detailed Tables, training curve Figures, and analysis are located in the github link.
>
> |Method|Acc(%)|
> |-|-|
> |FFA-LoRA |33.81|
> |FedMA |33.89|
> |Factor-wise|36.47|
> |FedProx |36.85|
> |**PERSIA** | **37.38**|
> |Full-rank|38.21|
>
> Results show:
> 1. Architecture generality: From encoder (RoBERTa-355M) to decoder (Qwen-3B)
> 2. Task generality: From classification to multi-step math reasoning
> 3. Consistency: PERSIA follows Full-rank closely.
>
> ---
> > **Q1:** *Practicality of federated FM fine-tuning*
>
> **Response Q1:** FM fine-tuning naturally suits better cross-silo FL (institutions, hospitals) instead of cross-device settings, with PEFT techniques like LoRA, which further enable resource-limited clients to participate, previously impractical.
>
> ---
> > **Q2:** *Concrete examples of GL(r) challenge*
>
> **Response Q2:** In Rank-1:
> $$B_1=[1,0]^\top,A_1=[1,0]\ \ and\ \ B_2=[-1,0]^\top,A_2=[-1,0]\ \ yield\ \ B_1A_1=B_2A_2,$$
> but factor-wise averaging gives $[0,0]$.
>
> Similarly,
> $$B_1=[1,0]^\top, A_1=[2,0]\ \ and\ \ B_2=[2,0]^\top, A_2=[1,0]$$
> share the same product, but averaging yields an incorrect result. With non-linear activations, these become *different functions* and generic FL handles them, but LoRA's linearity creates the GL($r$) ambiguity formalized in Corol. 3.5.
>
> ---
> ```
> [1] Guo, et al. "Selective aggregation for low-rank adaptation in federated learning." ICLR 2025.
> [2] Chen, et al. "Robust federated finetuning of LLMs via alternating optimization of LORA." NeurIPS 2025.
> ```

---

> > ### Author Rebuttal · Reviewer_aNKi · 2026-04-03
> >
> > I appreciate the author's clarification. I'll keep my score

---

> > > ### Author Response · Authors · 2026-04-07
> > >
> > > Thank you for the thorough engagement and for confirming that your concerns have been addressed.
> > >
> > > We will incorporate all clarifications into the camera-ready, including the **structured roadmap**, **Qwen2.5-3B** results, and **wall-clock analysis**.
> > > We appreciate your recognition of the $GL(r)$ insight and hope the additional experiments further demonstrate PERSIA's generality.

---

### Decision · Program_Chairs · 2026-04-30

**Decision:**

Reject

**Comment:**

While reviewers agreed that the paper studies an interesting and relevant problem in federated LoRA fine-tuning, and several reviewers found the core geometric intuition and proposed aggregation method promising, significant concerns remained about soundness of the paper’s claims, empirical support, and clarity of presentation.

The rebuttal addressed some concerns and added useful clarifications and experiments, and I have taken those responses into account. However, the discussion indicates that key concerns remained unresolved for multiple reviewers, especially regarding the practical significance of the gains, the adequacy of the experimental validation, and the clarity and precision of the theoretical presentation. Therefore, I do not recommend acceptance at this time. The problem is interesting and the approach has promise, but the paper would benefit from a clearer and more carefully scoped presentation together with stronger empirical validation.

Also, there is a incorrect reference in the paper: Wang, J., Yu, A. T., and Wornell, G. Federated learning with matched averaging. arXiv preprint arXiv:2002.06440, 2020.